# Improved Adversarial Diffusion Compression for Real-World Video Super-Resolution

**Bin Chen**[12*], **Weiqi Li**[12*], **Shijie Zhao**[2◇], **Xuanyu Zhang**[12], **Junlin Li**[2], **Li Zhang**[2], **Jian Zhang**[13†]

[1]School of Electronic and Computer Engineering, Peking University    [2]ByteDance Inc.
[3]Guangdong Provincial Key Laboratory of Ultra High Definition Immersive Media Technology, Shenzhen Graduate School, Peking University

## Abstract

While many diffusion models have achieved impressive results in real-world video super-resolution (Real-VSR) by generating rich and realistic details, their reliance on multi-step sampling leads to slow inference. One-step networks like SeedVR2, DOVE, and DLoRAL alleviate this through condensing generation into one single step, yet they remain heavy, with billions of parameters and multi-second latency. Recent adversarial diffusion compression (ADC) offers a promising path via pruning and distilling these models into a compact AdcSR network, but directly applying it to Real-VSR fails to balance spatial details and temporal consistency due to its lack of temporal awareness and the limitations of standard adversarial learning. To address these challenges, we propose an improved **ADC** method for Real-**VSR**. Our approach distills a large diffusion Transformer (DiT) teacher DOVE equipped with 3D spatio-temporal attentions, into a pruned 2D Stable Diffusion (SD)-based AdcSR backbone, augmented with lightweight 1D temporal convolutions, achieving significantly higher efficiency. In addition, we introduce a dual-head adversarial distillation scheme, in which discriminators in both pixel and feature domains explicitly disentangle the discrimination of details and consistency into two heads, enabling both objectives to be effectively optimized without sacrificing one for the other. Experiments demonstrate that the resulting compressed **AdcVSR** model reduces complexity by **95%** in parameters and achieves an **8×** acceleration over its DiT teacher DOVE, while maintaining competitive video quality and efficiency.

## 1 Introduction

Real-world video super-resolution (Real-VSR) (Tao et al., 2017; Nah et al., 2019) is a fundamental and long-standing problem in computer vision. It targets at recovering high-resolution (HR) videos $\mathbf{x}_{HR}$ from their low-resolution (LR) counterparts $\mathbf{x}_{LR}$ degraded by unknown factors in real-world cases. Traditional non-generative (Yi et al., 2019; Chan et al., 2021; Yang et al., 2021; Chan et al., 2022b;a; Wang et al., 2019; Liang et al., 2024) and generative adversarial network (GAN)-based (Chu et al., 2020; Lucas et al., 2019; Xu et al., 2025) approaches have achieved notable progress, yet most of them struggle under complex, mixed degradations, producing over-smoothed results or artifacts. To enhance the detail richness of super-resolution outputs $\mathbf{x}_{SR}$, many Real-VSR studies (Zhou et al., 2024; Yang et al., 2024a; He et al., 2024; Wang et al., 2025b; Xie et al., 2025b;a; Kong et al., 2025; Wang et al., 2025c;f; Zhao et al., 2025; Wang et al., 2025e; Bai et al., 2025) have already developed diffusion-based approaches which can generate video frames with richer and more realistic details. However, these methods are hindered by long inference time, as they need multiple sampling steps.

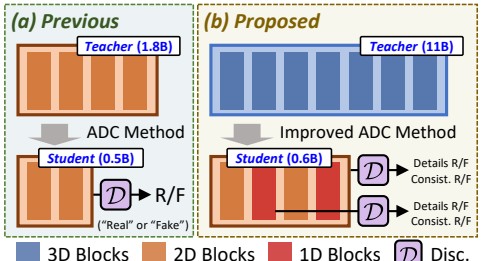

Figure 1: **Comparison of methods in compressing diffusion networks for Real-VSR.** **(a)** Traditional ADC (Chen et al., 2025a) distills an SD network with 2D spatial attentions into a pruned student using a single adversarial signal without temporal modeling, suffering from frame flickering. **(b)** Our improved ADC distills a larger DiT-based teacher with heavier 3D spatio-temporal attention into the same 2D student, augmented by 1D temporal convolutions, using dual-head discriminators $\mathcal{D}$ in pixel and feature domains. Through disentangling the discriminations of detail richness and temporal consistency into different heads, it balances the optimization of both.

*Equal contribution. ◇Project lead. †Corresponding author: zhangjian.sz@pku.edu.cn. This work was financially supported in part by National Natural Science Foundation of China (62372016), Guangdong Provincial Key Laboratory of Ultra High Definition Immersive Media Technology (2024B1212010006), Shenzhen Science and Technology Program (SYSPG20241211173440004), and Outstanding Talents Training Fund in Shenzhen.

Recently, researchers have shifted focus to one-step diffusion (Wang et al., 2024a; Wu et al., 2024; Xie et al., 2024; Lin et al., 2025c) for achieving efficient and high-quality Real-VSR. Building on the pretrained Stable Diffusion (SD) models (Rombach et al., 2022), originally developed for image generation, UltraVSR (Liu et al., 2025) enhances the temporal consistency in $\mathbf{x}_{SR}$ by propagating and fusing features along the temporal dimension, while DLoRAL (Sun et al., 2025) aligns the structure of the previous frame with the current one using the estimated optical flow between them. Building on video diffusion models, SeedVR2 (Wang et al., 2025a) progressively distills a pretrained 64-step SeedVR (Wang et al., 2025b) into one sampling step and further enhances it through adversarial post-training. DOVE (Chen et al., 2025b) adapts pretrained CogVideoX networks (Yang et al., 2024b) to Real-VSR by fine-tuning them on a curated high-quality video dataset. Despite these advances, such approaches still suffer from high complexity due to large-scale parameters and heavy computation.

Meanwhile, two recent methods, AdcSR (Chen et al., 2025a) and TinySR (Dong et al., 2025b), have explored compressing the diffusion networks OSEDiff (Wu et al., 2024) and TSD-SR (Dong et al., 2025a) via structural pruning and adversarial distillation to reduce complexity for real-world image super-resolution (Real-ISR), as shown in Fig. 1 (a). However, it is non-trivial to extend them to Real-VSR. One may use them to compress Real-VSR networks such as SeedVR2, DOVE, and DLoRAL, but two challenges arise: the compressed models are still too large, or video quality is compromised due to the conflicts between optimizing detail richness and temporal consistency (Sun et al., 2025), which are difficult to balance with existing distillation methods. Therefore, it remains underexplored and of great value to design more effective compression approaches for diffusion-based Real-VSR.

To improve effectiveness, in this paper, we propose **AdcVSR**, a novel Real-**VSR** network that compresses the one-step model DOVE using an improved method of **A**dversarial **D**iffusion **C**ompression (ADC) (Chen et al., 2025a). Unlike previous networks that rely on computationally costly 3D spatio-temporal attentions or additional frame alignment modules, we hypothesize that a 2D diffusion backbone (*e.g.*, SD2.1) is sufficient to generate rich details, while temporal consistency can be maintained with a few lightweight 1D temporal convolutional layers, and that their combination is also effective in removing degradations. Guided by this insight, as Fig. 1 (b) illustrates, AdcVSR adopts the same pruned SD2.1 backbone as AdcSR, augmented with 1D temporal convolution layers, achieving significantly lower complexity than its heavy 3D teacher DOVE. To improve video quality and address the conflict between optimizing details and consistency (Sun et al., 2025), we introduce a new adversarial distillation scheme that leverages the strong teacher DOVE together with numerous temporally consistent real videos and detail-rich real images. Specifically, two discriminators are employed for adversarial learning: one discriminating in a feature space of variational autoencoder (VAE) decoder and the other in pixel space, each with a "detail" head and a "consistency" head sharing a common backbone, to separately assess the realism of spatial details and temporal consistency. This enables the student to generate super-resolution results which are both detail-rich and temporally consistent. By integrating our "2D + 1D" architectural design with the dual-head, dual-discriminator adversarial distillation scheme, AdcVSR effectively compresses DOVE, yielding substantial efficiency gains while maintaining competitive video quality. Our contributions are summarized as follows:

❑ (1) We propose a novel improved ADC approach that combines an effective network design with adversarial distillation to compress a heavy Real-VSR model into an efficient diffusion-GAN hybrid.

❑ (2) We demonstrate that a 2D image diffusion backbone augmented with lightweight 1D temporal convolutions can effectively learn Real-VSR mapping from 3D diffusion Transformer (DiT) teacher.

❑ (3) We introduce a new adversarial distillation scheme that decouples the discriminations of details and consistency into two heads sharing a common network backbone, applied in both VAE decoder's feature space and the pixel space. This design enables balanced optimization, avoiding collapse into either over-smoothed outputs (loss of spatial details) or flickering (loss of temporal consistency).

❑ (4) Extensive experiments show that our AdcVSR model reduces parameters by 95% and achieves an $8\times$ acceleration over its teacher DOVE, while maintaining competitive performance on Real-VSR and striking a balance among fidelity, detail richness, temporal consistency, and model efficiency.

## 2 RELATED WORK

**Real-VSR.** A main challenge in this field lies in modeling the diverse, complex degradations of LR inputs, which can usually not be well represented by the bicubic downsampling (Mou et al., 2024;

Hu et al., 2023; Wang et al., 2022; Jiang et al., 2025; Li et al., 2025b). To address this, two strategies for collecting LR-HR video pairs to train deep Real-VSR networks have been developed: one captures pairs using different camera settings (Yang et al., 2021; Wang et al., 2023b), while the other synthesizes LR inputs from HR videos by composing degradations including noise, blur, resizing, and image/video coding compression (*e.g.*, JPEG, H.264, and MPEG-4) in random, high-order processes (Wang et al., 2021; Zhang et al., 2021; Chan et al., 2022b). These approaches enrich the degradation space and can synthesize large amounts of training data. Building upon these, a number of expressive Real-VSR networks have been developed (Shi et al., 2022). Non-generative methods like BasicVSR (Chan et al., 2021; 2022a), EDVR (Wang et al., 2019), and RVRT (Liang et al., 2024; 2022) excel at distortion removal, but often yield over-smoothed results under severe degradations. GAN-based methods including TecoGAN (Chu et al., 2020) and VideoGigaGAN (Xu et al., 2025) enhance visual quality with sharper details, but could introduce visible artifacts. Recently, diffusion-based methods have shown stronger performance by generating more realistic videos. For instance, Upscale-A-Video (Zhou et al., 2024), VEnhancer (He et al., 2024), STAR (Xie et al., 2025b), Real-isVSR (Zhao et al., 2025), SeTe-VSR (Wang et al., 2025e), and Vivid-VR (Bai et al., 2025) integrate transform/control modules into diffusion backbones to exploit the LR inputs as a condition. MGLD-VSR (Yang et al., 2024a) enforces temporal consistency via motion guidance, SeedVR (Wang et al., 2025b) introduces shifted window-based DiTs to handle varying spatial sizes, LiftVSR (Wang et al., 2025c) adopts attentions and memory for modeling short- and long-term dependencies, SimpleGVR (Xie et al., 2025a) performs cascaded latent-domain upscaling, and DiffVSR (Li et al., 2025c) decomposes Real-VSR learning in three progressive stages. Although these approaches improve detail richness, their reliance on multi-step sampling makes inference slow and computationally expensive.

**One-Step Diffusion.** Reducing step number while keeping output quality is a widely adopted strategy to accelerate inference (Wang et al., 2025f). In the context of Real-ISR and Real-VSR, several works have pushed this idea to the extreme by compressing multi-step generation into a single step (Yue et al., 2025; Gong et al., 2025; Wang et al., 2025d). For example, SinSR (Wang et al., 2024a) and SeedVR2 (Wang et al., 2025a) distill a 15-step ResShift (Yue et al., 2024) and 64-step SeedVR (Wang et al., 2025b) into one step through bidirectional or progressive distillation (Salimans & Ho, 2022). OSEDiff (Wu et al., 2024), S3Diff (Zhang et al., 2024a), D3SR (Li et al., 2024), and HYPIR (Lin et al., 2025c) adopt variational score distillation (Wang et al., 2024b), degradation-guided low-rank adaptations (LoRAs) (Hu et al., 2022), and adversarial post-training (Lin et al., 2025a;b), enabling one-step Real-ISR with high image quality. UltraVSR (Liu et al., 2025) aggregates temporal features via recurrent shifts, while PiSA-SR (Sun et al., 2024) and DLoRAL (Sun et al., 2025) introduce residual one-step diffusion and dual-LoRA learning, to alternately optimize spatial details and temporal consistency. Leveraging large-scale pretrained DiT-based text-to-image/-video (T2I/T2V) generation models, FluxSR (Li et al., 2025a), DiT4SR (Duan et al., 2025), and DOVE (Chen et al., 2025b) fine-tune FLUX (Labs, 2024), SD3 (Esser et al., 2024), and CogVideoX (Yang et al., 2024b) under the flow matching framework (Lipman et al., 2022) to enhance fine structures and text regions.

To further reduce complexity, AdcSR (Chen et al., 2025a), PassionSR (Zhu et al., 2025), and TinySR (Dong et al., 2025b) compress one-step Real-ISR networks using pruning, weight quantization, and distillation. However, these techniques are tailored for Real-ISR and struggle in Real-VSR, as they do not account for consistency. As a result, applying them directly would compromise video quality. To mitigate this problem, we propose to learn a diffusion network **AdcVSR** for Real-VSR, based on our improved ADC method that compresses the large DOVE teacher by distilling it into a pruned 2D SD2.1 backbone, augmented with lightweight 1D temporal convolutions. Additionally, we develop a new dual-head, dual-discriminator adversarial distillation scheme with decoupled detail-consistency discrimination, enabling the student network to achieve competitive video quality and efficiency.

# 3 METHOD

## 3.1 PRELIMINARY

**Conflict in Optimizing Details and Consistency.** Detail enrichment in video outputs requires synthesizing fine-grained structures like textures and edges with significant pixel-level variations, while temporal consistency demands constraining these variations across frames to ensure visually pleasant transitions and suppress flickering. These objectives are empirically found to be in conflict (Chu et al., 2020; Li et al., 2025c; Xu et al., 2025): many generative models emphasizing perceptual qual-

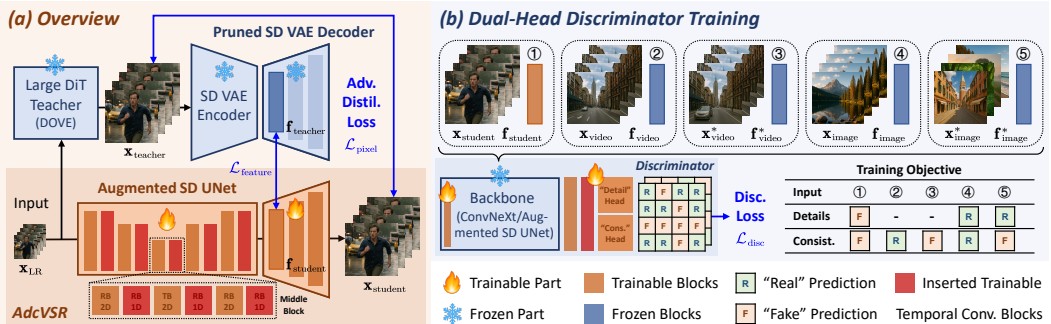

Figure 2: **Illustration of the proposed improved ADC method, with application to compressing DOVE (teacher) into AdcVSR model (student). (a)** We augment the 2D AdcSR Real-ISR network (Chen et al., 2025a), consisting of pruned SD2.1 UNet and VAE decoder, through inserting 1D temporal residual blocks (RBs) after each 2D spatial RB and Transformer block (TB), enabling temporal modeling, while maintaining efficiency for Real-VSR. The resulting AdcVSR network is then fully trained end-to-end via adversarial distillation from DOVE. **(b)** For adversarial learning, we design dual-head discriminators with pretrained backbones for feature extractions. Each discriminator uses 2D and 1D convolutions, followed by two linear projection heads at the tail, to separately evaluate detail richness and temporal consistency. Training is guided by five curated types of video and image data (1-5) with head-specific labels, achieving a balanced optimization of details and consistency.

ity tend to prioritize details, leading to visible flickers, while propagation or alignment mechanisms for consistency may over-smooth or attenuate details. Recent work (Sun et al., 2025) also highlights that details and consistency are competing objectives, where optimizing one could degrade the other.

**AdcSR, and Current Methods' Limitations.** AdcSR (Chen et al., 2025a) demonstrates that one-step Real-ISR diffusion network (Wu et al., 2024) can be compressed by an ADC method: removing VAE encoder and text components, pruning channels of denoising UNet and VAE decoder, and then applying adversarial distillation to restore outputs' quality, as shown in Fig. 1 (a). However, AdcSR is designed for Real-ISR and does not account for temporal modeling. When applied frame by frame to videos, it inevitably introduces flickers (Zhou et al., 2024; Rota et al., 2024). One-step Real-VSR networks (Liu et al., 2025; Wang et al., 2025a; Sun et al., 2025; Chen et al., 2025b) are still heavy, with $\geq$1.3B parameters and $\geq$4s latency even for a 25-frame $512 \times 512$ video (Fig. 4). An intuitive idea is to compress them by combining ADC, but existing learning approaches like dual-LoRA (Sun et al., 2024; 2025), adversarial/score-based distillation (Xu et al., 2025; Liu et al., 2025; Chen et al., 2025a) are ineffective under aggressive pruning, failing to resolve the detail-consistency conflict.

## 3.2 NETWORK ARCHITECTURE DESIGN

To compress large Real-VSR diffusion networks while balancing quality and efficiency, we propose an improved ADC approach which combines an effective architecture with an adversarial distillation scheme. Our key intuition is that although 3D spatio-temporal DiTs (Wang et al., 2025a; Chen et al., 2025b) achieve impressive results, their attention mechanisms aim to capture long-range space-time dependencies, which are important for T2V generation (Yang et al., 2024b; Wan et al., 2025), where such global information must be inferred from scratch. In Real-VSR, however, the LR video already provides much of this information (Sun et al., 2025), *e.g.*, structural layout and temporal continuity. Its main objectives are: **(1)** synthesizing details and **(2)** ensuring that they are temporally consistent to prevent flickering. In this setting, heavy 3D attentions might introduce redundancy, as much of its capacity is devoted to generating global spatio-temporal structures that are already present in $\mathbf{x}_{LR}$.

Building on this insight, we hypothesize that **(1)** a 2D diffusion backbone is capable of synthesizing details, and **(2)** consistency can be maintained with several 1D temporal convolutions. The rationale behind the second point is that maintaining consistency is inherently less challenging than synthesizing details: the objective is to constrain pixel-level variations across consecutive frames, rather than generate new fine structures. By enabling adjacent frames to be processed with temporal awareness, these convolutions are hypothesized to be sufficient to suppress flickering and yield temporally consistent recoveries. This motivates a principled "2D + 1D" network design that is expressive enough to learn the powerful Real-VSR mappings of large 3D DiTs while reducing redundant overhead.

Specifically, as Fig. 2 (a, bottom) exhibits, we adopt AdcSR (Chen et al., 2025a) as the 2D backbone, composed of channel-pruned SD2.1 UNet and VAE decoder. To add temporal awareness, we insert residual blocks after each UNet block, each consisting of a 1D temporal convolution, a ReLU activation, and a second convolution with a skip connection. Unlike flow- and motion-guided approaches (Zhou et al., 2024; Yang et al., 2024a) or alignment/propagation strategies (Liu et al., 2025; Xu et al., 2025; Sun et al., 2025), this simple yet effective design equips the resulting network, **AdcVSR**, with temporal modeling capacity, while keeping network architecture compact and inference efficient.

### 3.3 ADVERSARIAL DISTILLATION SCHEME

To achieve competitive video quality, we distill the large pretrained 3D DiT model DOVE (teacher) into our "2D + 1D" network (student). Specifically, as illustrated in Fig. 2 (a, top), we adopt DOVE's outputs as the learning target and conduct distillation in two domains: the pixel domain and the feature domain of the AdcSR VAE decoder's middle block. In the latter, DOVE's output pixels $\mathbf{x}_{\text{teacher}}$ are re-encoded by SD2.1 VAE encoder and fed into the middle block to obtain aligned features $\mathbf{f}_{\text{teacher}}$ for supervision, using the regression losses $\|\mathbf{x}_{\text{student}} - \mathbf{x}_{\text{teacher}}\|_1$ and $\|\mathbf{f}_{\text{student}} - \mathbf{f}_{\text{teacher}}\|_1$, where $\mathbf{x}_{\text{student}}$ and $\mathbf{f}_{\text{student}}$ denote the student's corresponding outputs. Compared with the original ADC framework (Chen et al., 2025a), which distills only in a single feature domain corresponding to $\mathbf{f}_{\text{student}}$ and $\mathbf{f}_{\text{teacher}}$ with the remaining decoder blocks frozen, our method uses richer supervisory signals and fine-tunes the entire network end-to-end, thereby activating its full capacity to learn Real-VSR mappings better.

However, although minimizing pixel and feature errors provides strong supervision, it is insufficient because our "2D + 1D" AdcVSR is significantly smaller and architecturally different from 3D DiT teacher, making exact fitting impractical while causing optimization difficulties and degraded reconstructions. Unlike the original setup of ADC, where the student was a streamlined variant of teacher with closer capacity and simple error minimization could still be effective, our case is far more challenging due to much larger gaps in both architecture design and parameter scale. We therefore retain error-minimizing distillations as a foundation, but augment them using adversarial learning to relax the requirement of exact replication, enabling the student to benefit from guidance of teacher DOVE while enjoying the flexibility to generate outputs that are feasible for its capacity and of high quality.

To achieve this, a straightforward approach would introduce a standard discriminator that adversarially aligns output distributions with real data (*e.g.*, $\mathbf{x}_{\text{HR}}$) (Sauer et al., 2023; 2024; Lin et al., 2024). However, this couples the objectives of details and consistency into one single adversarial signal. In practice, the discriminator $\mathcal{D}$ often tends to prioritize one aspect (typically details) at the expense of the other (typically consistency), leading to detail-rich but flickering result $\mathbf{x}_{\text{SR}}$. This reveals a fundamental issue: a traditional single-head discriminator entangles these conflicting objectives, yielding gradient that can not balance both. To overcome this, we propose a dual-head and dual-discriminator scheme that disentangles the assessments of details and consistency. Concretely, as Fig. 2 (b) shows, one discriminator operates in the pixel domain on $\mathbf{x}_{\text{student}}$, while the other in decoder feature domain on $\mathbf{f}_{\text{student}}$, forming a more comprehensive dual-domain supervision than single-domain approaches. Each discriminator is built upon a separate frozen pretrained backbone (ConvNeXt (Liu et al., 2022; Lin et al., 2025c) for $\mathbf{x}_{\text{student}}$ and the same augmented SD UNet as our designed AdcVSR for $\mathbf{f}_{\text{student}}$) to provide strong representations and stabilize training, followed by three additional alternating 2D and 1D convolutional layers to jointly capture spatial and temporal features. Finally, each discriminator branches into two linear heads ("detail" and "consistency") that project the last-layer features into two adversarial signals for detail realism and consistency, respectively. Formally, the adversarial distillation loss for the student generator is defined as follows, where we also include a perceptual DISTS (Ding et al., 2020) loss term as in DOVE to further strengthen the pixel-domain supervision:

$$\mathcal{L} = \lambda_{\text{pixel}}\mathcal{L}_{\text{pixel}} + \lambda_{\text{feature}}\mathcal{L}_{\text{feature}}, \tag{1}$$

$$\mathcal{L}_{\text{pixel}} = \|\mathbf{x}_{\text{student}} - \mathbf{x}_{\text{teacher}}\|_1 + \text{DISTS}(\mathbf{x}_{\text{student}}, \mathbf{x}_{\text{teacher}}) + \lambda_{\text{adv}}\text{Softplus}(-\mathcal{D}_{\text{pixel}}(\mathbf{x}_{\text{student}})), \tag{2}$$

$$\mathcal{L}_{\text{feature}} = \|\mathbf{f}_{\text{student}} - \mathbf{f}_{\text{teacher}}\|_1 + \lambda_{\text{adv}}\text{Softplus}(-\mathcal{D}_{\text{feature}}(\mathbf{f}_{\text{student}})), \tag{3}$$

where $\mathcal{D}_{\text{pixel}}$ and $\mathcal{D}_{\text{feature}}$ denote pixel- and feature-domain discriminators, $\text{Softplus}(-\mathcal{D}_{\text{pixel}}(\mathbf{x}_{\text{student}}))$ and $\text{Softplus}(-\mathcal{D}_{\text{feature}}(\mathbf{f}_{\text{student}}))$ implement non-saturating adversarial losses (Yin et al., 2024), while $\lambda_{\text{pixel}}$, $\lambda_{\text{feature}}$, and $\lambda_{\text{adv}}$ are weights controlling the relative contributions of corresponding loss terms.

To train the dual-head discriminators, we curate five carefully designed data types with head-specific labels that vary detail and consistency independently. First, the student's outputs ($\mathbf{x}_{\text{student}}$ and $\mathbf{f}_{\text{student}}$) are always labeled as "fake" for both heads, ensuring that adversarial feedback persistently pushes

the generator toward improvement. Second, real videos ($\mathbf{x}_\text{video}$ and $\mathbf{f}_\text{video}$) are adopted and labeled as "real" for consistency, as they preserve coherent temporal dynamics of the same underlying scene. Third, temporally shuffled versions of these videos ($\mathbf{x}_\text{video}^*$ and $\mathbf{f}_\text{video}^*$) obtained via randomly permuting frame order along the temporal dimension, destroy frame-to-frame continuity, and are therefore labeled as "fake" for consistency. Fourth, we exploit detail-rich real images by repeating each one to construct static pseudo-videos ($\mathbf{x}_\text{image}$ and $\mathbf{f}_\text{image}$), which enjoy both high-quality details and perfect temporal stability, and are thereby labeled as "real" for both heads. Finally, we randomly sample and crop real images without temporal correspondences ($\mathbf{x}_\text{image}^*$ and $\mathbf{f}_\text{image}^*$). These sequences are detail-rich but inherently inconsistent across frames, so they are labeled as "real" for details but "fake" for consistency. To be formal, the losses for dual-head discriminators $\mathcal{D}_\text{pixel}$ and $\mathcal{D}_\text{feature}$ are defined as:

$$\mathcal{L}_\text{disc} = \sum_{(\mathbf{s}, y_\text{d}, y_\text{c}) \in \mathcal{S}} \Big[ \text{Softplus}(-y_\text{d}[\mathcal{D}(\mathbf{s})]_\text{d}) + \text{Softplus}(-y_\text{c}[\mathcal{D}(\mathbf{s})]_\text{c}) \Big], \tag{4}$$

$$\mathcal{S} = \Big\{ (\mathbf{x}_\text{student}, -1, -1), (\mathbf{f}_\text{student}, -1, -1), (\mathbf{x}_\text{video}, 0, 1), (\mathbf{f}_\text{video}, 0, 1), (\mathbf{x}_\text{video}^*, 0, -1),$$
$$(\mathbf{f}_\text{video}^*, 0, -1), (\mathbf{x}_\text{image}, 1, 1), (\mathbf{f}_\text{image}, 1, 1), (\mathbf{x}_\text{image}^*, 1, -1), (\mathbf{f}_\text{image}^*, 1, -1) \Big\}. \tag{5}$$

Here, $[\mathcal{D}(\mathbf{s})]_\text{d}$ and $[\mathcal{D}(\mathbf{s})]_\text{c}$ denote the outputs of the "detail" and "consistency" heads of discriminator $\mathcal{D}$ for input $\mathbf{s}$, while $y_\text{d}, y_\text{c} \in \{-1, 0, 1\}$ encode "fake", "unlabeled", and "real" labels, respectively. The set $\mathcal{S}$ enumerates the five curated data types and corresponding labels in both pixel and feature domains. It is worth noting that we leave real video details unlabeled, and rely on real images as the positive supervision for "detail" head, encouraging the generator to produce more detail-rich frames.

In contrast to standard GAN discriminators (Goodfellow et al., 2014) which provide a single binary signal for real versus generated samples, our design restructures adversarial supervision into a multi-attribute form, producing fine-grained and disentangled signals for both details and consistency, with outputs that preserve the same spatial resolution as the inputs. This approach moves beyond existing adversarial distillation methods (Sauer et al., 2023; 2024; Chen et al., 2025a) by explicitly requiring the dual-head discriminators to evaluate two aspects of video realism. As a result, neither aspect can be disregarded or down-weighted, as the two dedicated heads consistently provide supervisions, ensuring that the model receives separate weight gradients for both. This prevents AdcVSR generator from collapsing toward one objective at the expense of the other, instead guiding it to optimize both jointly, and produce reconstructions that are simultaneously detail-rich and temporally consistent.

## 4 EXPERIMENT

### 4.1 EXPERIMENTAL SETTING

**Implementation Details.** We employ AdcSR (Chen et al., 2025a) pretrained by compressing PiSA-SR (Sun et al., 2024) as 2D backbone, in which the SD2.1 denoising UNet and VAE decoder (Luo et al., 2023) are channel-pruned by 25% and 50%, respectively, and augmented with zero-initialized (Zhang et al., 2023) 1D temporal convolutions to form our AdcVSR network. Each 1D convolution has a kernel size of 3 and the same channel number as its preceding UNet block. For discriminators, the channel numbers of first convolutions are adjusted to match the dimensions of input images and features, while both channel numbers of last-layer features are set to 256. The "detail" and "consistency" heads are implemented by $1 \times 1$ convolutions with 192 and 64 output channels, respectively.

Similar to Wang et al. (2021), AdcVSR model is trained in two consecutive stages. In the first stage, we perform only error-minimizing distillation from the pretrained DOVE teacher without adversarial learning for 200K iterations. In the second stage, AdcVSR (generator) is initialized with the weights from the first stage and fine-tuned for another 200K iterations. During this stage, the pixel-domain discriminator uses the pretrained ConvNeXt backbone from the OpenCLIP library (kept frozen) (Liu et al., 2022; Lin et al., 2025c), while the feature-domain discriminator exploits the same pretrained augmented SD UNet from the first stage (also frozen). The initial learning rates for AdcVSR are set to $1 \times 10^{-4}$ in the first stage and $1 \times 10^{-5}$ in the second stage, each halved after 100K iterations, while the trainable parts of discriminators (first and tail convolutions, as well as heads) adopt learning rate $1 \times 10^{-7}$. Loss weights are set to $\lambda_\text{pixel} = 0.1$, $\lambda_\text{feature} = 1.0$, and $\lambda_\text{adv} = 1.0$, respectively.

In both stages, we fully fine-tune the entire AdcVSR network end-to-end, following Liu et al. (2025); Sun et al. (2025); Chen et al. (2025b), using randomly sampled and cropped temporally consistent

Table 1: **Quantitative comparison of Real-VSR performance.** Inference time is measured on an NVIDIA H20 GPU for generating a 25-frame video at spatial resolution $512\times512$. The best, second-best, and third-best results are labeled in **bold red**, underlined blue, and *italic green*, respectively.

| Method | RealBasicVSR | Upscale-A-Video | MGLD-VSR | STAR | SeedVR2 | DOVE | DLoRAL | PiSA-SR | AdcSR | HYPIR | AdcVSR (Ours) |
|---|---|---|---|---|---|---|---|---|---|---|---|
| | | | | | Test Dataset: UDM10 (Synthetic) | | | | | | |
| PSNR↑ | 24.39 | 23.03 | 24.28 | 22.61 | 25.92 | **26.00** | 22.49 | 23.21 | 23.39 | 22.55 | *25.36* |
| SSIM↑ | 0.7376 | 0.6189 | 0.7491 | 0.6534 | *0.7674* | **0.7805** | 0.7130 | 0.6799 | 0.6772 | 0.6995 | 0.7697 |
| LPIPS↓ | 0.3283 | 0.4218 | 0.3103 | 0.5055 | 0.2653 | **0.2645** | 0.3201 | 0.3658 | 0.3781 | 0.3736 | *0.3065* |
| DISTS↓ | 0.2078 | 0.2360 | *0.1909* | 0.2665 | **0.1532** | 0.1732 | 0.2066 | 0.2213 | 0.2287 | 0.2125 | 0.2112 |
| MANIQA↑ | 0.5725 | 0.5331 | 0.5558 | 0.3468 | 0.5232 | 0.5133 | 0.5679 | **0.6257** | 0.5696 | 0.5856 | *0.5793* |
| CLIPIQA↑ | 0.4422 | 0.4661 | 0.4640 | 0.2346 | 0.3471 | 0.5420 | 0.4667 | **0.7055** | *0.6693* | 0.6006 | 0.6818 |
| MUSIQ↑ | 57.10 | 52.06 | 58.12 | 33.93 | 50.09 | 60.68 | 55.54 | **66.42** | *61.30* | 59.85 | 63.88 |
| $E^*_{\text{warp}}$↓ | 3.36 | 3.68 | 3.31 | *2.51* | 2.56 | 2.22 | 3.51 | 6.96 | 6.19 | 10.68 | **1.67** |
| DOVER↑ | 0.2610 | 0.4002 | 0.3311 | 0.2717 | 0.3296 | 0.4731 | 0.3637 | **0.5010** | 0.4364 | *0.4851* | 0.4878 |
| | | | | | Test Dataset: VideoLQ (Real-World) | | | | | | |
| MANIQA↑ | 0.5609 | 0.5366 | 0.5530 | 0.4356 | 0.4389 | 0.4336 | 0.5976 | 0.6319 | 0.6017 | **0.6424** | *0.6121* |
| CLIPIQA↑ | 0.3444 | 0.3594 | 0.3446 | 0.2497 | 0.2318 | 0.3258 | 0.4211 | **0.6199** | 0.6098 | 0.5937 | *0.6024* |
| MUSIQ↑ | 56.47 | 55.20 | 55.90 | 41.01 | 40.56 | 50.03 | 58.50 | **67.31** | 66.14 | 63.69 | *64.55* |
| $E^*_{\text{warp}}$↓ | 9.27 | 14.54 | 8.99 | 10.65 | 11.32 | 8.41 | *8.94* | 12.65 | 12.47 | 23.45 | **6.74** |
| DOVER↑ | 0.2239 | 0.3278 | 0.2830 | 0.3013 | 0.2027 | 0.3790 | 0.3192 | *0.4131* | 0.4100 | **0.4711** | 0.4319 |
| | | | | | Efficiency | | | | | | |
| #Steps↓ | - | *30* | 50 | 15 | 1 | 1 | 1 | 1 | 1 | 1 | 1 |
| #Param. (B)↓ | **0.04** | 14.44 | 1.43 | 2.49 | 8.24 | 10.55 | 1.30 | 1.30 | 0.46 | 1.55 | *0.57* |
| Time (s)↓ | **0.35** | 66.39 | 32.34 | 96.38 | 60.61 | 4.42 | 6.36 | 2.94 | 0.52 | 2.81 | *0.55* |

real video and detail-rich real image data from OpenVid-1M (Nan et al., 2024) and LSDIR (Li et al., 2023). We use a batch size of 8, with 25 frames per video clip and a spatial resolution of $512 \times 512$. The RealBasicVSR degradation pipeline (Chan et al., 2022b) is applied to synthesize LR-HR video pairs. All experiments are implemented in PyTorch, and trained with the Adam optimizer (Kingma, 2014) on 8 NVIDIA H20 GPUs (96GB each), with the full training process taking about one day.

**Test Datasets.** Following Liu et al. (2025); Wang et al. (2025a); Sun et al. (2025), we test AdcVSR and compare it with other methods using the same synthetic and real-world datasets as DOVE (Chen et al., 2025b). The three synthetic test datasets include UDM10 (Yi et al., 2019) (10 videos), SPMCS (Tao et al., 2017) (30 videos), and YouHQ40 (Zhou et al., 2024) (40 videos), which are synthesized with the same degradation pipeline as during training, using a scaling factor of 4 for Real-VSR task. The three real-world datasets are RealVSR (Yang et al., 2021) (50 videos), MVSR4x (Wang et al., 2023b) (15 videos), and VideoLQ (Chan et al., 2022b) (50 videos). All videos are pre-processed via clipping the first 25 frames and applying center-cropping, fixing the output spatial size to $512 \times 512$.

**Evaluation Metrics.** We employ both full-reference and no-reference metrics for performance evaluations. Full-reference metrics include PSNR and SSIM (Wang et al., 2004) for fidelity, as well as LPIPS (Zhang et al., 2018) and DISTS (Ding et al., 2020) for perceptual quality. No-reference metrics include MANIQA (Yang et al., 2022), CLIPIQA (Wang et al., 2023a), and MUSIQ (Ke et al., 2021). In addition, following Yang et al. (2024a); Zhang et al. (2024b); Sun et al. (2025), we report the flow warping error $E^*_{\text{warp}}$ (Lai et al., 2018), scaled by $10^{-3}$, as in DOVE (Chen et al., 2025b), and employ DOVER (Wu et al., 2023), to evaluate temporal consistency and video quality, respectively.

## 4.2 COMPARISON WITH STATE-OF-THE-ARTS

**Compared Methods.** Following Liu et al. (2025); Wang et al. (2025a); Sun et al. (2025); Chen et al. (2025b), we compare the proposed AdcVSR model with seven representative Real-VSR approaches: the non-generative RealBasicVSR (Chan et al., 2022b); multi-step diffusion-based Upscale-A-Video (Zhou et al., 2024), MGLD-VSR (Yang et al., 2024a), and STAR (Xie et al., 2025b); as well as one-step diffusion networks SeedVR2 (Wang et al., 2025a), DOVE (Chen et al., 2025b), and DLoRAL (Sun et al., 2025). Additionally, we include three state-of-the-art one-step diffusion-based Real-ISR approaches: PiSA-SR (Sun et al., 2024), AdcSR (Chen et al., 2025a), and HYPIR (Lin et al., 2025c), which super-resolve video frames independently, for comprehensive evaluation and comparison.

**Video Quality Comparison.** Tab. 1 quantitatively demonstrates that our AdcVSR achieves competitive performance across a broad range of metrics. First, it ranks within the top three in most cases, surpassing the majority of competing approaches and confirming its effectiveness in restoring high-quality video frames. Second, AdcVSR achieves strong temporal consistency with smallest warping errors, as indicated by its superior $E^*_{\text{warp}}$ results. In contrast, Real-ISR models perform the worst on this metric because they lack temporal modeling, which leads to inconsistent content across frames. Third, when compared with the previous best approaches DOVE (teacher) and PiSA-SR in fidelity, perceptual quality, and warping error, AdcVSR remains highly competitive across all these aspects.

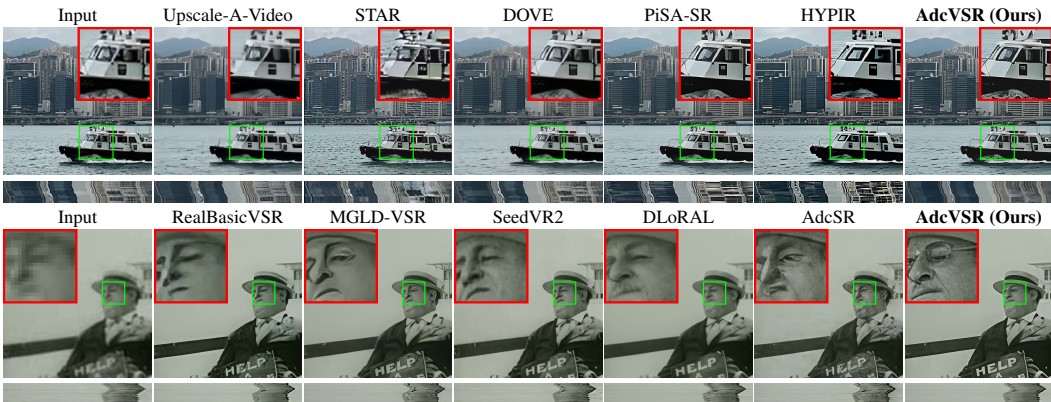

Figure 3: **Qualitative comparison of Real-VSR performance** on 13th frames of two videos: "028" from RealVSR (top) and "016" from VideoLQ (bottom). Temporal profiles are provided below each frame, obtained by taking slices along the width-temporal plane at the vertical centers of the frames.

The advantages of AdcVSR are also verified by the qualitative comparison in Fig. 3. Our network reconstructs sharp and realistic details, while RealBasicVSR, Upscale-A-Video, MGLD-VSR, DOVE, and DLoRAL yield over-smoothed outputs. STAR and AdcSR bring artifacts to boat's windows and facial regions, whereas PiSA-SR and HYPIR produce fewer details on building and water surface, or generate visually unrealistic boat textures. Moreover, they suffer from significant temporal instability, as evidenced by the erratic fluctuations in their temporal profiles, resulting in unpleasant flickers that degrade overall video quality. In comparison, our AdcVSR not only reconstructs natural details on buildings, boat structures, water textures, facial components, clothing, hat, and signage with less distortion, but also maintains smooth transitions across consecutive frames with reduced flickering.

It is worth highlighting from Tab. 1 and Fig. 3 that, despite their poor temporal consistency, Real-ISR diffusion networks PiSA-SR, AdcSR, and HYPIR, with only 2D spatial attentions and convolutions, are highly effective at removing degradations in individual video frames and generating rich details. This results in high-quality outputs with strong scores on no-reference perceptual metrics, including MANIQA, CLIPIQA, MUSIQ, and DOVER, which often align better with human perception than traditional metrics like PSNR. This observation is consistent with hypothesis **(1)** in Sec. 3.2. Building on this insight, our method employs lightweight temporal convolutions together with dual-head adversarial distillation, allowing the preservation of strong per-frame detail quality while improving fidelity and temporal consistency across all frames.

**Efficiency Comparison.** The final 4 rows of Tab. 1 and the bubble plot in Fig. 4 compare Real-VSR performance in temporal consistency ($E^*_{\mathrm{warp}}$), step numbers, parameter numbers, and inference times across different approaches. Built upon an effective "2D + 1D" architecture, and learned from the large 3D DiT teacher DOVE using our dual-head adversarial distillation scheme, AdcVSR delivers both strong temporal consistency with the best $E^*_{\mathrm{warp}}$ results and efficiency gains. Specifically, it reduces parameters by 96%, 60%, and 77%, and accelerates inference by 121×, 59×, and 175× over the multi-step diffusion-based methods Upscale-A-Video, MGLD-VSR, and STAR. Against one-step diffusion-based Real-VSR models SeedVR2 and DLoRAL, it achieves parameter reductions of 93% and 56%, and accelerations of 110× and 308×, respectively. Compared with its teacher DOVE, AdcVSR achieves **a 95% reduction** in parameters and **an 8× speedup** while maintaining very competitive video quality. Overall, AdcVSR is substantially faster and more lightweight than most existing approaches, verifying the effectiveness of our improved ADC method and the high efficiency of the resulting compressed diffusion network.

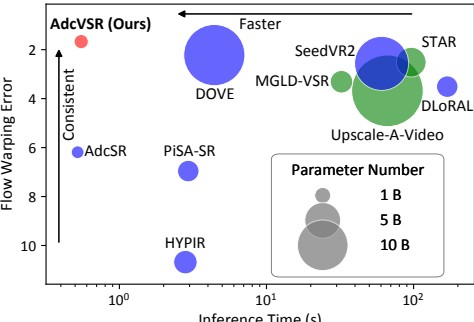

Figure 4: **Performance comparison among diffusion-based Real-VSR methods** in temporal consistency and complexity (parameter number and inference time) (see Tab. 1). AdcVSR attains the lowest warping error $E^*_{\mathrm{warp}}$, the second-lowest parameter number, and the second-highest inference speed. Bubble colors represent method types: green for multi-step, blue for one-step, and red for AdcVSR.

## 4.3 ABLATION STUDY

**Effect of "2D + 1D" Network Design.** Tab. 2 and Fig. 5 compare three student architectures: a pruned 3D DiT (based on DOVE) obtained by the original ADC approach, a 2D SD backbone (AdcSR), and our AdcVSR. The 3D DiT model delivers the best DISTS and strong $E^*_{\text{warp}}$ but remains heavy. Replacing it with a 2D backbone severely degrades performance, showing that a model without temporal modeling cannot effectively learn from a temporally aware teacher, or maintain both frame quality and inter-frame co-herence, leading to visible flickering. By introducing 1D convolutions, our design restores temporal modeling capacity while preserving efficiency: it achieves the lowest $E^*_{\text{warp}}$, narrows DISTS gap to 3D model to 0.0014 with 7% parameters, and produces sharp textures and smooth temporal profiles.

Table 2: **Comparison of network designs** on UDM10.

| Method | DISTS↓ | $E^*_{\text{warp}}$↓ | #Param. (B)↓ |
|---|---|---|---|
| 3D (A Pruned DOVE) | **0.2098** | 2.53 | 8.36 |
| 2D (AdcSR) | 0.2418 | 4.43 | **0.52** |
| **2D + 1D (Ours)** | 0.2112 | **1.67** | 0.55 |

Figure 5: **Comparison of different network designs**.

**Effect of Dual-Head Discriminators.** Tab. 3 compares three AdcVSR variants with different settings for the discriminators: (1) single-head (only one output without distinguishing details and consistencies), (2) single-domain

Table 3: **Comparison of discriminators** on YouHQ40.

| Method | CLIPIQA↑ | $E^*_{\text{warp}}$↓ |
|---|---|---|
| Single-Head, Dual-Domain | 0.6745 | 6.32 |
| Dual-Head, Single-Domain | 0.6421 | 3.59 |
| **Dual-Head, Dual-Domain (Ours)** | **0.6861** | **2.22** |

(only one feature-domain discriminator as in the original ADC), and (3) our proposed scheme. The single-head variant preserves frame quality but shows much worse $E^*_{\text{warp}}$, indicating that consistency is less optimized during its adversarial training. The single-domain variant improves consistency but reduces perceptual quality due to the absence of pixel-domain supervision. In contrast, our method achieves best performance on both metrics, effectively balancing details and temporal consistency.

**Effect of Adversarial Distillation.** In Tab. 4, we compare AdcVSR variants under different setups of adversarial learning and distillation. Removing adversarial losses or relying solely on ground truth (GT) supervisions noticeably degrades LPIPS and MUSIQ, indicating that both adversarial training and a teacher's guid-

Table 4: **Comparison of distillation setups** on MVSR4x.

| Method | PSNR↑ | LPIPS↓ | MUSIQ↑ |
|---|---|---|---|
| No Adversarial Loss | 23.97 | 0.3596 | 54.33 |
| No Teacher (HR GT Only) | **24.85** | 0.3641 | 50.32 |
| SeedVR2 as Teacher | 23.24 | 0.3489 | 60.74 |
| DLoRAL as Teacher | 23.08 | 0.3554 | 54.61 |
| **DOVE as Teacher (Ours)** | 23.81 | **0.3337** | **61.48** |

ance are essential for perceptual quality. Using SeedVR2 or DLoRAL as teachers yields promising but weaker results. By contrast, our adversarial distillation with DOVE as teacher strikes a favorable balance across three metrics, demonstrating that adversarial learning combined with an appropriate teacher is important for improving Real-VSR performance in both fidelity and perceptual realism.

Due to page limitations, more experimental results, analyses, and discussions are presented in the **Appendix**.

## 5 CONCLUSION

In this work, we proposed an improved **A**dversarial **D**iffusion **C**ompression (ADC) method for real-world **V**ideo **S**uper-**R**esolution (Real-VSR). Instead of relying on computationally heavy 3D spatio-temporal attentions as in existing diffusion Transformer (DiT)-based approaches, our model adopted a compact "2D + 1D" design: a pruned 2D Stable Diffusion (SD) backbone for synthesizing details, augmented with lightweight 1D temporal convolutions to enforce inter-frame coherence, while their combination also proved effective in removing degradations. To address the conflicts between optimizing detail richness and temporal consistency in Real-VSR while leveraging the knowledge of a large 3D DiT teacher DOVE as well as diverse real video and image data, we introduced a dual-head, dual-discriminator adversarial distillation scheme that disentangles and jointly optimizes details and consistency via pixel- and feature-domain supervision. Across synthetic and real-world benchmarks, the resulting **AdcVSR** model achieved competitive video quality while being substantially more efficient than its 3D DiT teacher, offering a **95%** parameter reduction and an **8×** inference acceleration, striking a strong balance among fidelity, detail richness, temporal consistency, and model efficiency. Beyond Real-VSR, our work provides a systematic recipe for building efficient video reconstruction systems, delivering practical guidelines for diffusion model compression and real-world application.

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

APPENDIX

Our main paper outlines the core ideas and techniques of the proposed approach, demonstrating the effectiveness of our main methodological contributions, including (1) the "2D + 1D" network architecture design and (2) dual-head adversarial distillation scheme, via experimental validation. In this **Appendix**, we provide additional information, including the training pseudocode of our improved ADC method in Sec. A, more ablation studies in Sec. B, additional comparison results in Sec. C, discussion on limitations in Sec. D, and statement of large language model (LLM) usage in Sec. E.

## A    TRAINING PSEUDOCODE OF ADCVSR

The training pseudocode of AdcVSR is outlined in Algo. 1, and consists of two consecutive stages, as explained in Sec. 4.1. In Stage 1, we augment the pretrained AdcSR model by adding 1D residual blocks (RBs), each containing two temporal convolutions, to create AdcVSR. We then perform knowledge distillation from the large 3D DiT teacher DOVE without adversarial learning, by minimizing the errors between the student's and teacher's outputs in both the pixel and feature domains. In Stage 2, we initialize the student model with the pretrained version from Stage 1, and incorporate both the pixel- and feature-domain discriminators. Adversarial losses in both domains further refine the student's ability to generate rich, realistic details and maintain temporal consistency. This stage allows the model to effectively learn from DOVE while achieving a balance between detail richness and temporal consistency, thus addressing the conflict between optimizing these two objectives.

---

**Algorithm 1:** Training of Our Proposed AdcVSR

---

**Input:** Pretrained 3D DiT DOVE, AdcSR, and SD2.1 models; Weights $\lambda_{\text{pixel}}$, $\lambda_{\text{feature}}$, and $\lambda_{\text{adv}}$.

**Stage 1: Knowledge Distillation Without Adversarial Learning**
Add 1D temporal convolutions to AdcSR as described in Sec. 3.2 to obtain AdcVSR network;
**for** *number of training iterations* **do**
> Sample a batch of LR-HR video pairs $(\mathbf{x}_{\text{LR}}, \mathbf{x}_{\text{HR}})$;
> Compute the outputs $\mathbf{x}_{\text{teacher}}$ and $\mathbf{f}_{\text{teacher}}$ from DOVE (teacher);
> Compute the outputs $\mathbf{x}_{\text{student}}$ and $\mathbf{f}_{\text{student}}$ from AdcVSR (student);
> Compute the distillation loss:
> $\mathcal{L}_{\text{distil}} = \lambda_{\text{pixel}} \left[ \|\mathbf{x}_{\text{student}} - \mathbf{x}_{\text{teacher}}\|_1 + \text{DISTS}(\mathbf{x}_{\text{student}}, \mathbf{x}_{\text{teacher}}) \right] + \lambda_{\text{feature}} \|\mathbf{f}_{\text{student}} - \mathbf{f}_{\text{teacher}}\|_1$;
> Update the model weights of AdcVSR (student) using the Adam optimizer;

**Stage 2: Dual-Head, Dual-Discriminator Adversarial Distillation**
Initialize AdcVSR (student, generator) from the pretrained model in Stage 1;
Initialize pixel- and feature-domain discriminators $\mathcal{D}_{\text{pixel}}$ and $\mathcal{D}_{\text{feature}}$ as described in Sec. 4.1;
**for** *number of training iterations* **do**
> Sample a batch of LR-HR video pairs $(\mathbf{x}_{\text{LR}}, \mathbf{x}_{\text{HR}})$;
> Compute the outputs $\mathbf{x}_{\text{teacher}}$ and $\mathbf{f}_{\text{teacher}}$ from DOVE (teacher);
> Compute the outputs $\mathbf{x}_{\text{student}}$ and $\mathbf{f}_{\text{student}}$ from AdcVSR (student);
> Compute the distillation loss as in Eq. (1);
> Update the model weights of AdcVSR (student) using the Adam optimizer;
> Sample three batches of video and image data: $\mathbf{x}_{\text{video}}$, $\mathbf{x}_{\text{image}}$, and $\mathbf{x}_{\text{image}}^*$;
> Construct a batch of pseudo-videos $\mathbf{x}_{\text{video}}^*$ by randomly permuting the frames within each
>   individual video of $\mathbf{x}_{\text{video}}$, ensuring that each real video is shuffled independently;
> Compute features $\mathbf{f}_{\text{video}}$, $\mathbf{f}_{\text{video}}^*$, $\mathbf{f}_{\text{image}}$, and $\mathbf{f}_{\text{image}}^*$ using $\mathbf{x}_{\text{video}}$, $\mathbf{x}_{\text{video}}^*$, $\mathbf{x}_{\text{image}}$, and $\mathbf{x}_{\text{image}}^*$;
> Compute losses for discriminators $\mathcal{D}_{\text{pixel}}$ and $\mathcal{D}_{\text{feature}}$ as in Eq. (4);
> Update the model weights of discriminators using the Adam optimizer;

---

# B    MORE ABLATION STUDIES

**Effect of "2D + 1D" Network Design.** Tab. 5 and Fig. 6 compare four students: (1) a 2D SD backbone (AdcSR) without temporal modeling, same as in Tab. 2, (2) our AdcVSR with one 1D residual block (RB) after each UNet block, (3) a variant with doubled 1D convolutions, and (4) a variant replacing the 1D convolutions with temporal self-attention modules (Zhou et al., 2024). The 2D network suffers from large performance drops: without temporal receptive fields, distillation becomes difficult and both temporal consistency and frames' quality degrade heavily, as reflected in its poor metrics and higher loss values. Adding 1D temporal convolutions notably reduces the loss to a lower plateau and improves video quality by providing the network with the ability to capture temporal dependencies. Doubling 1D convolutions brings no clear gains, in-

Table 5: **Comparison of network designs** on SPMCS.

| Method | LPIPS↓ | MANIQA↑ | $E_{\text{warp}}^*$↓ |
|---|---|---|---|
| 2D (AdcSR) | 0.3625 | 0.6120 | 5.84 |
| **2D + 1D Temp. Conv. (Ours)** | 0.3282 | **0.6500** | **1.59** |
| 2D + 1D Temp. Conv. Doubled | **0.3215** | 0.6462 | 1.68 |
| 2D + 1D Temporal Attention | 0.3391 | 0.6384 | 2.08 |

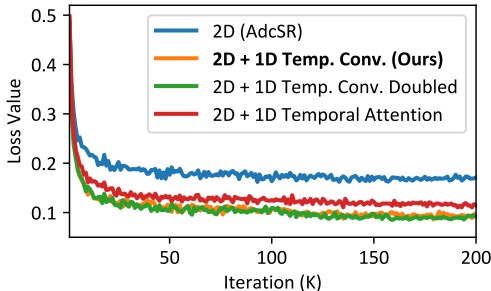

Figure 6: **Comparison of distillation loss** in Stage 1.

dicating that the main factor for student performance is the acquisition of temporal receptive fields rather than simply increasing parameter number or depth. Replacing 1D convolutions with temporal attentions does not improve quality or efficiency, and makes optimization more difficult, as shown by higher converged loss. Overall, our design achieves strong LPIPS, MANIQA, and $E_{\text{warp}}^*$ results with minor overhead, verifying that adding several 1D temporal convolutions to a 2D backbone yields an effective student capable of learning from a 3D DiT teacher while remaining compact and efficient.

**Effect of Dual-Head Discriminators.** To examine how allocating channels between the two heads influences outputs' quality, we vary their split under a fixed total of 256 channels across five settings, as presented in Tab. 6 and Fig. 7. At one extreme (100% / 0%), the setup degenerates to a single-head detail discriminator, which achieves the best MUSIQ result with sharp details but fails to maintain temporal consistency, yielding the worst $E_{\text{warp}}^*$ and low DOVER, leading to visible flickers in temporal profiles (*e.g.*, in the upper branches, building facades, stripes, and lower vegetation). At the opposite extreme (0% / 100%), the model suppresses inter-frame

Table 6: **Comparison of dual-head splits** on RealVSR.

| Head Split (Detail / Consistency) | MUSIQ↑ | $E_{\text{warp}}^*$↓ | DOVER↑ |
|---|---|---|---|
| 100% / 0% (256 / 0 Channels) | **73.10** | 8.85 | 0.4520 |
| **75% / 25% (192 / 64 Ch., Ours)** | 72.95 | 3.28 | **0.4875** |
| 50% / 50% (128 / 128 Channels) | 70.80 | 3.22 | 0.4802 |
| 25% / 75% (64 / 192 Channels) | 68.39 | **3.15** | 0.4631 |
| 0% / 100% (0 / 256 Channels) | 65.21 | 3.18 | 0.4410 |

100% / 0%  **75% / 25%**  50% / 50%  25% / 75%  0% / 100%

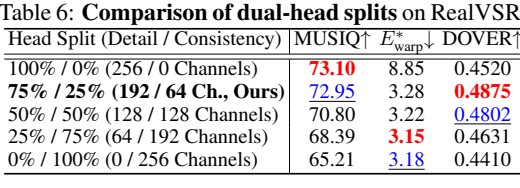
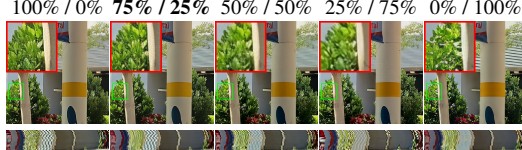

Figure 7: **Comparison of various head channel splits**.

variation, slightly reducing $E_{\text{warp}}^*$ but at the significant expense of perceptual fidelity, producing over-smoothed textures and artifacts. Our default split (75% / 25%) strikes a strong balance: it achieves the highest DOVER while maintaining high MUSIQ and low $E_{\text{warp}}^*$, showing that dedicating a modest portion of capacity to "consistency" is sufficient when both heads share the same backbone and output layers. Intermediate splits (50% / 50%, or 25% / 75%) do not clearly improve performance, yielding lower MUSIQ and DOVER with few leaf details. Overall, these results confirm that channel allocation between the two heads is critical, and that assigning a minority share (25%) to consistency delivers balanced optimization of detail richness and temporal consistency, validating our design.

Additionally, to validate whether sharing backbones and tail convolutions is necessary for two heads in $\mathcal{D}$, we compare our dual-head design (shared backbones and tail convolutions for the two heads) against a variant employing separate single-head discriminators for both details and consistency (no sharing). In the latter case, the "consistency" branch produces unstable loss values and the training collapses. We conjecture this is because shuffling frames ($\mathbf{x}_{\text{video}}^*$) or assembling different images ($\mathbf{x}_{\text{image}}^*$) as "fake" videos may not provide sufficient supervision for independent consistency discriminators. In contrast, forcing two heads to share the same backbone and tail convolutions stabilizes training: the last-layer features are compelled to encode both detail realism and temporal coherence, allowing the consistency head to judge temporal flickering by leveraging detail information. In this way, our design introduces an implicit regularization, making the optimization of consistency stable.

Table 8: **Quantitative comparison of Real-VSR performance** on 4 additional benchmark datasets.

| Method | RealBasicVSR | Upscale-A-Video | MGLD-VSR | STAR | SeedVR2 | DOVE | DLoRAL | PiSA-SR | AdcSR | HYPIR | AdcVSR (Ours) |
|---|---|---|---|---|---|---|---|---|---|---|---|
| | | | | | Test Dataset: SPMCS (Synthetic) | | | | | | |
| PSNR↑ | 20.69 | 19.83 | *20.83* | 20.32 | 20.89 | **20.94** | 20.35 | 20.12 | 20.14 | 18.13 | *20.83* |
| SSIM↑ | 0.5056 | 0.4301 | *0.5201* | 0.4897 | **0.5234** | 0.5221 | 0.4955 | 0.4674 | 0.4702 | 0.4284 | 0.5055 |
| LPIPS↓ | 0.3886 | 0.4336 | 0.3583 | 0.5657 | **0.3122** | 0.3165 | 0.3531 | 0.3662 | 0.3695 | 0.3969 | *0.3282* |
| DISTS↓ | 0.2334 | 0.2466 | 0.2198 | 0.2957 | **0.1823** | 0.1896 | 0.2164 | 0.2286 | 0.2304 | 0.2251 | *0.2036* |
| MANIQA↑ | 0.5977 | 0.5862 | 0.5941 | 0.3654 | 0.6289 | 0.5774 | 0.6066 | 0.6561 | 0.6332 | **0.6961** | *0.6500* |
| CLIPIQA↑ | 0.4563 | 0.4892 | 0.4358 | 0.2663 | 0.4764 | 0.6142 | 0.4917 | 0.6971 | 0.6716 | **0.7099** | *0.6962* |
| MUSIQ↑ | 64.88 | 66.50 | 64.96 | 34.76 | 63.89 | 68.41 | 63.67 | **70.61** | 69.78 | 70.23 | *69.94* |
| $E^*_{warp}$↓ | 1.73 | 2.81 | 1.83 | 1.10 | *1.38* | **0.98** | 2.76 | 5.30 | 5.52 | 13.33 | 1.59 |
| DOVER↑ | 0.2792 | 0.3834 | 0.3326 | 0.2333 | 0.3769 | 0.4759 | 0.3691 | 0.4445 | *0.4638* | **0.5383** | 0.4622 |
| | | | | | Test Dataset: YouHQ40 (Synthetic) | | | | | | |
| PSNR↑ | 21.74 | 21.08 | 22.38 | 21.35 | 22.70 | **23.44** | 20.85 | 21.53 | 21.51 | 19.40 | 23.31 |
| SSIM↑ | 0.5491 | 0.4817 | 0.5679 | 0.5462 | *0.5847* | **0.6068** | 0.5322 | 0.5166 | 0.5204 | 0.4782 | 0.5998 |
| LPIPS↓ | 0.4039 | 0.4137 | 0.3796 | 0.5195 | **0.2839** | 0.3319 | *0.3454* | 0.3632 | 0.3603 | 0.3903 | 0.3539 |
| DISTS↓ | 0.2324 | 0.2200 | 0.2199 | 0.2627 | **0.1604** | 0.1935 | *0.1955* | 0.2164 | 0.2071 | 0.2163 | 0.2126 |
| MANIQA↑ | 0.5860 | 0.6122 | 0.5860 | 0.3622 | 0.6212 | 0.5058 | 0.6072 | 0.6551 | 0.6330 | **0.6718** | 0.6587 |
| CLIPIQA↑ | 0.4557 | 0.4896 | 0.4343 | 0.2650 | 0.4156 | 0.5117 | 0.4698 | **0.6973** | 0.6921 | *0.6870* | 0.6861 |
| MUSIQ↑ | 62.14 | 63.42 | 61.97 | 33.48 | 61.47 | 59.56 | 59.68 | **69.41** | 67.97 | 69.12 | *68.28* |
| $E^*_{warp}$↓ | 4.30 | 6.20 | 4.41 | 2.38 | 5.30 | *2.63* | 4.51 | 7.27 | 7.39 | 17.88 | **2.22** |
| DOVER↑ | 0.3507 | 0.5691 | 0.4288 | 0.3767 | 0.5011 | 0.5649 | 0.5015 | 0.5930 | *0.5993* | **0.6393** | 0.6098 |
| | | | | | Test Dataset: RealVSR (Real-World) | | | | | | |
| PSNR↑ | 21.98 | 20.47 | 21.44 | 18.07 | 20.75 | **22.05** | 18.46 | 19.87 | 19.27 | 16.52 | *21.53* |
| SSIM↑ | 0.7113 | 0.6005 | 0.6450 | 0.5273 | *0.6970* | **0.7234** | 0.5284 | 0.6113 | 0.5634 | 0.4928 | 0.6668 |
| LPIPS↓ | *0.1899* | 0.2540 | 0.2439 | 0.3180 | 0.1810 | **0.1703** | *0.1133* | 0.2092 | 0.2590 | 0.2975 | 0.1920 |
| DISTS↓ | 0.1143 | 0.1489 | 0.1499 | 0.1742 | **0.1080** | 0.1111 | *0.1133* | 0.1511 | 0.1657 | 0.2019 | 0.1364 |
| MANIQA↑ | 0.6755 | 0.6278 | 0.6479 | 0.6431 | 0.6746 | 0.6769 | 0.6837 | 0.7374 | 0.6655 | **0.7671** | *0.6939* |
| CLIPIQA↑ | 0.3418 | 0.4340 | 0.3273 | 0.3289 | 0.3137 | 0.5394 | 0.4331 | **0.6822** | 0.6581 | 0.6704 | *0.6646* |
| MUSIQ↑ | 68.13 | 67.96 | 67.18 | 63.80 | 64.61 | 71.55 | 70.36 | **73.83** | 72.49 | 73.07 | 72.95 |
| $E^*_{warp}$↓ | *3.60* | 5.06 | **3.18** | 7.16 | 4.21 | 3.82 | 5.30 | 8.97 | 13.78 | 22.82 | 3.28 |
| DOVER↑ | 0.3482 | 0.3420 | 0.3353 | 0.4063 | 0.4028 | *0.5284* | 0.3789 | 0.5377 | 0.4578 | **0.5711** | 0.4875 |
| | | | | | Test Dataset: MVSR4x (Real-World) | | | | | | |
| PSNR↑ | 21.89 | 21.30 | 22.55 | 21.90 | *22.18* | 22.15 | 21.87 | 21.95 | 22.03 | 20.81 | **23.81** |
| SSIM↑ | 0.7181 | 0.6662 | 0.7325 | *0.7346* | 0.7597 | 0.7229 | 0.7292 | 0.7182 | 0.7115 | 0.7017 | **0.7750** |
| LPIPS↓ | 0.3816 | 0.4161 | 0.3578 | 0.4529 | 0.3419 | 0.3643 | *0.3487* | 0.3727 | 0.3960 | 0.4133 | **0.3337** |
| DISTS↓ | 0.2498 | 0.2657 | **0.2446** | 0.2781 | 0.2453 | 0.2500 | 0.2488 | 0.2638 | 0.2756 | 0.2716 | 0.2537 |
| MANIQA↑ | 0.5378 | 0.5316 | 0.4930 | 0.2829 | 0.3265 | 0.4973 | 0.4983 | 0.5231 | *0.5367* | **0.5470** | 0.5283 |
| CLIPIQA↑ | 0.4729 | 0.4938 | 0.3221 | 0.2299 | 0.2258 | 0.5453 | 0.4560 | 0.6086 | **0.6527** | 0.5807 | *0.5903* |
| MUSIQ↑ | 58.35 | 61.34 | 54.07 | 24.76 | 29.14 | 62.22 | 51.47 | 61.27 | **62.46** | 56.05 | *61.48* |
| $E^*_{warp}$↓ | 1.64 | 3.10 | 1.94 | **0.78** | *0.98* | 1.01 | 1.60 | 3.02 | 3.12 | 4.54 | 0.79 |
| DOVER↑ | 0.2455 | 0.3652 | 0.2428 | 0.0951 | 0.0831 | 0.3811 | 0.2586 | 0.3684 | *0.3824* | **0.4006** | 0.3860 |

**Effect of Curated Data for Dual-Head Training.** We ablate the five curated data types used to train the dual-head discriminators, and investigate whether they are necessary to disentangle details' richness from temporal consistency. As shown in Tab. 7, first, training only with videos

Table 7: **Ablation of curated data for** $\mathcal{D}$ **on UDM10.**

| Training Configuration | LPIPS↓ | CLIPIQA↑ | $E^*_{warp}$↓ |
|---|---|---|---|
| No $\mathbf{x}^*_{video}$ and No $\mathbf{x}^*_{image}$ (Types 3 & 5) | 0.3430 | 0.6521 | 5.92 |
| Real Videos as Detail-Real (Label "1") | 0.3152 | 0.6703 | 1.98 |
| Teacher Videos Replacing Real Videos | 0.3224 | 0.6652 | 2.47 |
| **Full Scheme (Ours)** | **0.3065** | **0.6818** | **1.67** |

$\mathbf{x}_{video}$ and images $\mathbf{x}_{image}$ (*i.e.*, removing the shuffled videos $\mathbf{x}^*_{video}$ and the assembled image sequences $\mathbf{x}^*_{image}$) fails to penalize flicker: $E^*_{warp}$ increases and LPIPS worsens, indicating that the heads' objectives are entangled and optimization deteriorates. Second, if we label real video details as positives for the detail head, temporal consistency remains decent and $E^*_{warp}$ is strong, but perceptual quality drops. This might stem from that most real videos in OpenVid-1M are not as high-quality or detail-rich as the images of LSDIR, causing the model to align with lower-quality frames. Third, replacing real videos with teacher outputs $\mathbf{x}_{teacher}$ and $\mathbf{f}_{teacher}$ as "real" for the consistency head degrades both perceptual quality and consistency, with worse $E^*_{warp}$. This may be due to that teacher super-resolved results lack real, natural temporal dynamics of real-world videos. In contrast, our scheme combines teacher outputs with real images/videos to provide complementary supervision signals. This leads to the lowest LPIPS, the highest CLIPIQA, and the best $E^*_{warp}$, confirming that our curated data for the two heads are essential to avoid collapse and to balance the optimization of details and consistency.

# C  MORE COMPARISON RESULTS

**Quantitative Comparison.** In Tab. 8, we present a comprehensive comparison of AdcVSR against ten state-of-the-art approaches (same as in the main paper), evaluated with nine metrics across four test datasets: SPMCS, YouHQ40, RealVSR, and MVSR4x. Consistent with the results in Tab. 1, AdcVSR ranks within the top-3 in 24 out of 36 cases (two-thirds overall), a higher proportion than other competing approaches, demonstrating its competitive Real-VSR performance. While advanced one-step diffusion-based Real-VSR models (SeedVR2, DOVE, DLoRAL) and leading Real-ISR models

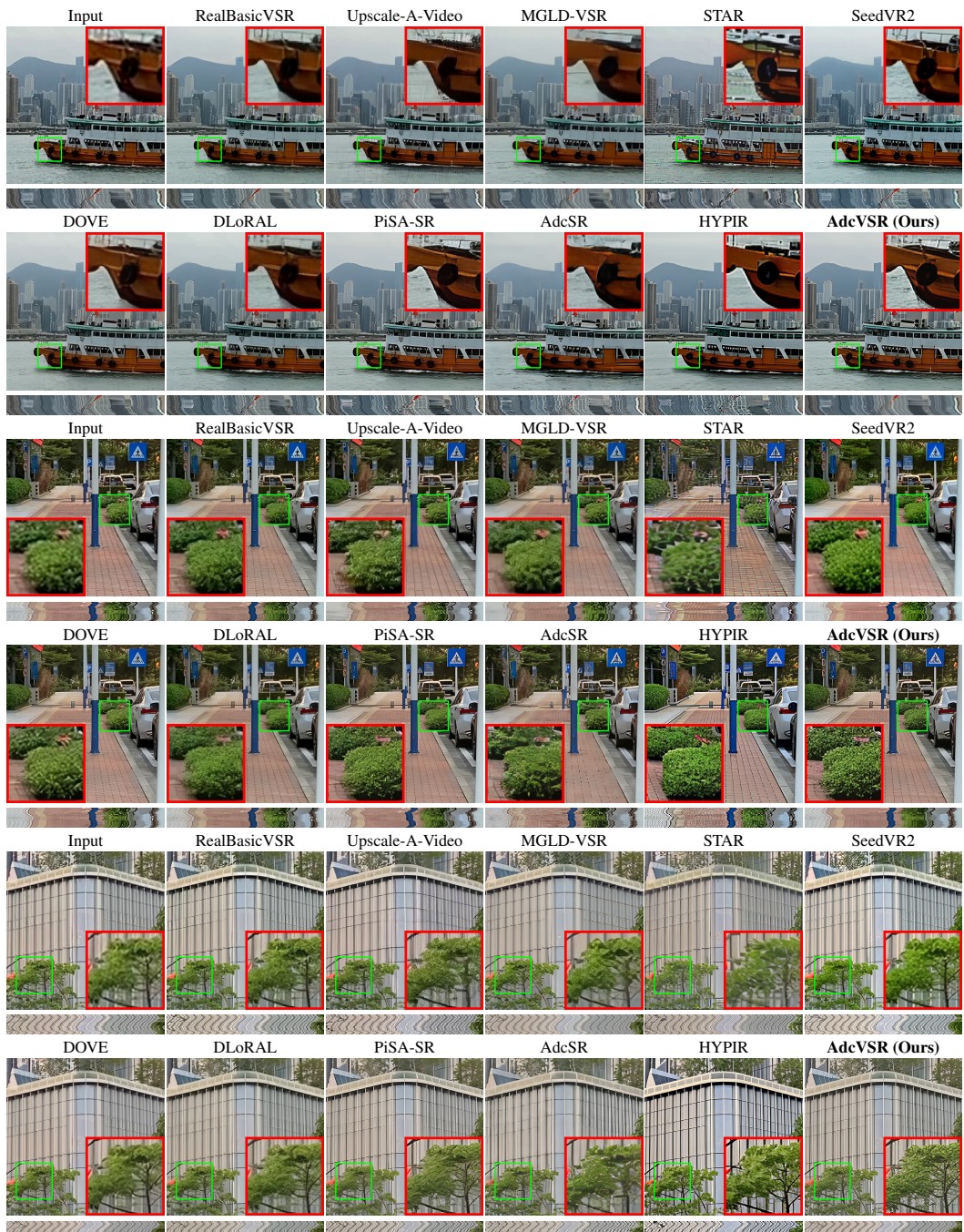

Figure 8: **Qualitative comparison of Real-VSR performance** on three real videos: the 4th frame of "029" (top), the 2nd frame of "090" (middle), and the 15th frame of "247" (bottom) from RealVSR.

(PiSA-SR, AdcSR, HYPIR) perform well, they tend to be biased toward either fidelity or perceptual quality but rarely both. In comparison, AdcVSR not only achieves many top-3 and top-2 results but also several times secures the best (top-1) scores, exhibiting a favorable balance across fidelity, perceptual quality, and temporal consistency, delivering both high-quality video frames (PSNR, SSIM, LPIPS, DISTS, MANIQA, CLIPIQA, MUSIQ), and strong temporal consistency ($E_{\text{warp}}^*$), as well as overall video quality (DOVER). Importantly, compared with its teacher networks DOVE and PiSA-SR, AdcVSR attains comparable fidelity and perceptual quality while maintaining competitive temporal consistency. Note that it is much more efficient, requiring only 5% and 44% of the parameters, while providing 8× and 5× faster inferences over DOVE and PiSA-SR, respectively. These results

validate that AdcVSR is both effective and efficient, confirming the effectiveness of the "2D + 1D" architecture design and dual-head adversarial distillation scheme in our improved ADC method.

**Qualitative Comparison.** Fig. 8 shows visual comparisons across super-resolved videos produced by different approaches. We observe that RealBasicVSR, MGLD-VSR, SeedVR2, DOVE, and DLo-RAL often fail to recover sufficient fine details, resulting in over-smoothed or blurry textures, as seen in the orange hull region of the ferry and in the foliage areas. Methods including Upscale-A-Video, STAR, and AdcSR tend to introduce noticeable artifacts, particularly in high-frequency regions like tree leaves and the ship bow, leading to unnatural appearances. PiSA-SR and HYPIR produce relatively sharper results but sometimes generate unrealistic or inconsistent details, occasionally accompanied by shifts in brightness compared with the input. In contrast, our AdcVSR produces sharp and realistic details with a balanced trade-off between fidelity and perceptual quality, while avoiding the artifacts and distortions observed in other approaches. Furthermore, the temporal consistency of our results remains stable across consecutive frames, as reflected in its smooth temporal profiles without significant flickering. Overall, these results are consistent with our quantitative findings, validating the ability of AdcVSR to generate rich and fine visual details while preserving temporal coherence.

## D    LIMITATIONS

Despite the effectiveness of proposed AdcVSR, several limitations remain. First, while the approach produces rich and sharp details, certain challenging structures (*e.g.*, fine textures such as foliage, water surfaces, or transparent regions such as glass) are still not faithfully reconstructed. In these cases, AdcVSR may generate softened details, or partially hallucinated patterns. Second, the performance of our model is inherently tied to the base SD2.1 backbone and its teacher networks. Although our improved ADC method effectively compresses them, the capacity gap between student and teacher can still limit final reconstruction fidelity, particularly under complex motion or severe degradations. Third, the reliance on curated video and image datasets together with adversarial learning makes the model sensitive to training data quality and might introduce instability during optimization. Finally, while our design strikes a strong balance between fidelity, perceptual quality, and temporal consistency, the trade-offs are not perfect, and minor flickering may still occur under extreme degradations.

**Future Work.** In the future, we plan to leverage stronger generative priors from larger networks, investigate more advanced temporal modeling strategies, and refine the adversarial distillation scheme to further enhance reconstruction performance and robustness. These directions may help resolve the above limitations and extend the applicability of our method to more diverse real-world scenarios.

## E    LLM USAGE STATEMENT

We used a large language model (LLM) only for minor grammar and phrasing polishes. All technical content, including ideas, experiments, analyses, and discussions, was entirely created by the authors.

