# OpenReview forum: "Improved Adversarial Diffusion Compression for Real-World Video Super-Resolution"
_ICLR.cc/2026/Conference — ICLR 2026 Poster_

### Official Review · Reviewer_c8ag · 2025-10-15

**Soundness:** 2
**Presentation:** 2
**Contribution:** 2
**Rating:** 4
**Confidence:** 5

**Summary:**

This paper proposed an improved adversarial diffusion compression method for real-world Video Super-Resolution. It comprises a 2D SD backbone and several 1D convs to replace the heavy 3D DiT architecture to improves its efficiency. By using a dual-head discriminator, it balances the optimization between detail enhancement and temporal consistency for adversarial distillation scheme. Experiments on synthetic and real-world benchmarks show its potential in compressing Real-VSR model on both model size and inference steps effectively.

**Strengths:**

* This paper proposes an intuitive explanation to use a ``2D + 1D'' architecture for student model to improve its efficiency.
* Detailed designs on adversarial distillation scheme (e.g., data type, training loss) help improve its clarity.
* Thorough experiments on synthetic and real-world benchmarks demonstrate its effectiveness after distillation.

**Weaknesses:**

* I remain skeptical about the effectiveness of replacing the 3D DiT architecture with a ``2D+1D'' architecture. As demonstrated in the paper, it uses a radical distillation strategy to achieve a 20-fold reduction in parameter count and single-step inference. It is essential to demonstrate that this strategy does not cause significant performance degradation on real-world benchmarks. Ablation study is only conducted in UDM10 (a small synthetic benchmark). Authors should conduct more analysis on large-scale real-world benchmarks.
* Detailed investigation on model size is missing. This paper provides an appealing results by distilling a 0.6B student model from a 11B teacher model. However, the selection of model size requires further exploration. Authors need to find a boundary between visual quality and model size, and conduct an ablation study about it.
* Motivation of using dual-domain discriminator. As mentioned in Sec 3.3, it replaces the conventional feat-domain discriminator by two discriminators in both pixel and feat domain. However, the motivation of adding one discrimintor in pixel domain remains unclear. Some related questions remain unresolved, such as why two discriminators are not used in the feature domain, and whether using more discriminators would improve performance.
* Authors should also conduct experiments about temporal quality. By using 1D convs and a relative small model size, the temporal consistency of AdcVSR remians unclear now. Authors should conduct experiments by showing multiple frames of a single video in real-world benchmark.

**Questions:**

* Using stronger generative backbone. Using SD2.1 as 2D backbone limits the capacity on complex degradations. Authors should consider utilizing a large-scale backbone.
* I strongly recommend authors to provide a video demo or video files corresponding to the results presented in the paper. Given the usage of a ``2D+1D'' architecture, verifying the temporal quality of the final results is essential. Displaying only a single frame from the video within the article is not very convincing. I will consider raising my score after evaluating the video results.

---

> ### Author Response · Authors · 2025-11-21
> **Author Response (1/n)**
>
> > **Q1: More analyses on real-world benchmark regarding the use of "2D + 1D" architecture.**
>
> A1: Thank you for your insightful comment and valuable suggestion. Following your recommendation, and to more comprehensively verify the effectiveness of replacing the 3D DiT (pruned using the original AdcSR method) with a "2D + 1D" architecture, we conducted additional experiments of more metrics on the larger-scale real-world benchmark RealVSR, in addition to the original ablation study on UDM10. The results are shown below:
>
> Comparison of network designs on UDM10 and RealVSR.
>
> | Method          | PSNR↑ | SSIM↑  | LPIPS↓ | DISTS↓ | $E_{\text{warp}}^{*}$↓ | #Param.↓ |
> |-----------------|--------|---------|---------|---------|--------------------------|---------|
> | 3D              | **25.51** | 0.7648  | **0.2988** | **0.2098** | 2.53                     | 8.36B   |
> | **2D + 1D (Ours)** | 25.36 | **0.7697** | 0.3065  | 0.2112  | **1.67**                 | **0.55B** |
>
> | Method          | PSNR↑ | SSIM↑ | LPIPS↓ | DISTS↓ | $E_{\text{warp}}^{*}$↓ | #Param.↓ |
> |-----------------|--------|--------|---------|---------|--------------------------|---------|
> | 3D              | **21.75** | **0.6912** | **0.1865** | **0.1273** | 3.79                     | 8.36B   |
> | **2D + 1D (Ours)** | 21.53 | 0.6668 | 0.1920  | 0.1364  | **3.28**                 | **0.55B** |
>
> From these results, we observe that the "2D + 1D" architecture maintains competitive performance. To be specific, it achieves scores close to the 3D model, with PSNR difference <0.3 dB, SSIM difference <0.03, LPIPS difference <0.01, DISTS difference <0.01, and better temporal consistency ($E_{\text{warp}}^{*}$), while reducing parameters by 93% (from 8.36B to 0.55B). These findings demonstrate that this strategy does not introduce significant performance degradation on real-world benchmark.
>
> We hope that these additional experiments and clarifications can help resolve your doubt, and make the effectiveness clearer.
>
> ---
>
> > **Q2: Ablation study on model size.**
>
> A2: Thank you for your constructive suggestion. Following your recommendation, we conducted additional experiments to investigate how the model size affects video quality on the real-world benchmark RealVSR. Concretely, we vary the channel-pruning ratio of the SD2 UNet in AdcSR to control the corresponding AdcVSR model size (the default setting uses 25% channel pruning, same as the original AdcSR). The results are provided below:
>
> | Channel Pruning Ratio for SD2 UNet | PSNR↑ | SSIM↑ | LPIPS↓ | DISTS↓ | $E_{\text{warp}}^{\*}$↓ | #Param.↓ |
> |------------------------------------|--------|--------|---------|---------|---------------------------|---------|
> | 0%                                 | **21.65**  | **0.6760**  | **0.1895**  | **0.1320**   | 3.41                      | 0.90B   |
> | **25% (Ours)**                     | 21.53  | 0.6668 | 0.1920   | 0.1364  | **3.28**                  | 0.55B   |
> | 50%                                | 21.05  | 0.6412 | 0.2088  | 0.1518  | 4.05                      | 0.29B   |
> | 75%                                | 20.47  | 0.6031 | 0.3345  | 0.2837  | 6.32                      | **0.14B** |
>
> From these results, we observe a trade-off between visual quality and efficiency. Increasing the pruning ratio from 25% to 50% and 75%, which reduces the model size from 0.55B to 0.29B and 0.14B, leads to a significant drop in reconstruction quality and temporal consistency due to insufficient model capacity. In contrast, reducing the pruning ratio to 0% enlarges the model to 0.90B and yields a modest quality improvement. Overall, the default 25% configuration provides a balanced operating point that maintains competitive video quality while keeping the model compact. This supports our default choice of a 0.6B student model as an effective and efficient design.
>
> We hope that these additional experiments and analyses can help address your concern, and make the impact of model size on performance clearer.

---

> ### Author Response · Authors · 2025-11-21
> **Author Response (2/n)**
>
> > **Q3: Motivation for using dual-domain discriminators.**
>
> A3: Thank you for raising this important point. As stated in the third paragraph of Section 3.3 (the text highlighted in blue), **the use of two discriminators in both feature and pixel domains is intended to provide the generator with richer and more diverse supervision signals than single-domain variants.** Compared to previous methods like AdcSR, which rely solely on a feature-domain discriminator, adding an additional discriminator in the pixel domain encourages the generator to match not only high-level VAE decoder feature statistics but also low-level pixel realism. This complementary supervision strengthens training and leads to higher-quality video outputs.
>
> Regarding why we do not use more discriminators within the same domain, we find that doing so does not further improve performance. Once the generator (student) is already supervised by two complementary domains, additional discriminators in a single domain may tend to introduce redundant or highly correlated gradients rather than genuinely new constraints. In practice, this redundancy increases training computational costs without offering notable video quality gains, causing performance to saturate rather than improve. This is why our design emphasizes complementarity across domains (feature + pixel) rather than simply increasing the number of discriminators.
>
> To verify this, we further conducted an extended ablation study on the YouHQ40 dataset to examine how reducing or increasing the number of discriminators affects video quality. The results are shown below:
>
> | Method                                                       | CLIPIQA↑ | $E_{\text{warp}}^{\*}$↓ |
> |-------------------------------------------------------------|-----------|--------------------------|
> | one discriminator in feature domain                         | 0.6421    | 3.59                     |
> | one discriminator in pixel domain                           | 0.6209    | 2.87                     |
> | **one in feature domain, one in pixel domain (Ours)**       | **0.6861**| 2.22                 |
> | two in feature domain, one in pixel domain                  | 0.6837    | **2.14**                     |
> | one in feature domain, two in pixel domain                  | 0.6845    | 2.36                     |
>
> From these results, we observe that using only one discriminator causes a clear degradation in video quality metrics CLIPIQA and $E_{\text{warp}}^{\*}$ because the generator receives weaker supervision and fewer constraints. In contrast, adding more discriminators within the same domain does not yield notable improvements: performance remains similar, while training time increases substantially. These findings support our choice of the dual-domain design.
>
> We hope that these additional clarifications and experiments can help make the motivation and necessity of dual-domain discriminators clearer.
>
> ---
>
> > **Q4: Show multiple frames of a single video in real-world benchmark.**
>
> A4: Thank you for pointing this out. We agree that providing multiple frames is helpful for assessing temporal consistency. Following your recommendation, we have additionally uploaded supplementary material (included in the newly added zip file) containing a demo of real-world VideoLQ sequence, where we display multiple frames for the low-resolution (LR) input, baseline AdcSR, teacher DOVE, and our AdcVSR outputs.
>
> From these visual results, we observe that inserting 1D temporal convolutions and applying adversarial distillation noticeably improves temporal stability: AdcVSR produces frames that are more temporally consistent across time, whereas the original model (AdcSR) without 1D temporal convolutions exhibits stronger flickering. This helps clarify the practical effect of our design on real-world videos.
>
> Thank you again for the helpful suggestion. We hope these additional results can address your concern and provide a clearer understanding of AdcVSR's temporal consistency.

---

> ### Author Response · Authors · 2025-11-21
> **Author Response (3/3)**
>
> > **Q5: Use a larger-scale backbone.**
>
> A5: Thank you for your constructive suggestion. Following your recommendation, we conducted additional experiments using a larger-scale backbone SDXL (https://huggingface.co/stabilityai/stable-diffusion-xl-base-1.0) in place of the originally adopted smaller SD2.1, while keeping the rest of our method framework unchanged. The results on YouHQ40 and RealVSR are reported below:
>
> | Backbone      | PSNR↑ | SSIM↑ | LPIPS↓ | DISTS↓ | $E_{\text{warp}}^{*}$↓ | #Param.↓ |
> |---------------|--------|--------|----------|-----------|---------------------------|---------|
> | **SDXL**       | **23.39** | **0.6027** | **0.3476** | **0.2072** | **2.18** | 1.91B |
> | **SD2.1 (Ours)** | 23.31 | 0.5998 | 0.3539 | 0.2126 | 2.22 | **0.55B** |
>
> | Backbone      | PSNR↑ | SSIM↑ | LPIPS↓ | DISTS↓ | $E_{\text{warp}}^{*}$↓ | #Param.↓ |
> |---------------|--------|--------|----------|-----------|---------------------------|---------|
> | **SDXL**       | **21.70** | **0.6864** | **0.1887** | **0.1294** | **3.16** | 1.91B |
> | **SD2.1 (Ours)** | 21.53 | 0.6668 | 0.1920 | 0.1364 | 3.28 | **0.55B** |
>
> From these results, we observe that adopting a larger-scale backbone can further improve reconstruction quality. Specifically, replacing SD2.1 with SDXL yields moderate but consistent gains in fidelity and perceptual quality across both datasets. This improvement, however, comes at the cost of significantly increased parameters (from 0.55B to 1.91B, more than 3×). These findings suggest that a model with larger capacity indeed helps handle more complex degradations, as you pointed out, but also introduces a notable increase in model size and cost.
>
> We hope that these additional experiments can help address your doubt, and clarify the impact of adopting a larger-scale backbone.
>
> ---
>
> > **Q6: Provide a video demo or files.**
>
> A6: Thank you for your kind and constructive suggestion. We agree that providing a demo can make the method's performance clearer. Following your recommendation, we have uploaded supplementary material (the newly added zip file) containing a demo video corresponding to the real-world VideoLQ benchmark (sequence "016"), consistent with the example shown in the paper. Along with our AdcVSR's output, we also include the low-resolution (LR) input, the original image diffusion SR network AdcSR's output, and the teacher model DOVE's output for comparison. For completeness, we additionally provide the decomposed frame sequences for reference.
>
> From these visual results, we observe that AdcVSR achieves a balance between detail richness and temporal consistency. Compared with the teacher model DOVE, it produces vivid and sharper details; and compared with the original image SR network AdcSR, it generates more temporally consistent frames. This is due to the introduced temporal convolution modules and our improved adversarial diffusion compression and distillation strategy.
>
> We hope these additional results can help resolve your doubt, and further demonstrate the effectiveness of our method.

---

> > ### Comment · Reviewer_c8ag · 2025-11-27
> > **Official Response by Reviewer c8ag**
> >
> > Thank you for the detailed clarifications and the additional experiments. I believe your responses adequately address the main concerns raised in my earlier review. After further consideration, I am willing to raise my score to 6 in recognition of the improvements and supplementary results provided.

---

> > > ### Author Response · Authors · 2025-11-28
> > >
> > > Thank you for your follow-up message and for taking the time to review our additional analyses and experiments. We are glad that our clarifications addressed your concerns, and we sincerely appreciate your decision to raise the score. Your detailed feedback has been highly valuable in strengthening the paper.

---

### Official Review · Reviewer_7Ea8 · 2025-10-30

**Soundness:** 4
**Presentation:** 3
**Contribution:** 4
**Rating:** 6
**Confidence:** 3

**Summary:**

This paper proposes an improved adversarial diffusion compression method for real-world video super-resolution tasks. The authors note that while existing diffusion-based VSR methods can generate detail-rich videos, their inference speed is slow; conversely, one-step approaches are faster but still result in bulky models. To address this, the authors introduce AdcVSR, which distills a DiT teacher model with 3D spatiotemporal attention into a lightweight "2D + 1D" student network (based on a trimmed Stable Diffusion 2.1 backbone combined with 1D temporal convolutions). Additionally, a dual-head dual-discriminator adversarial distillation strategy is introduced to decouple the discrimination of detail richness and temporal consistency in the pixel and feature domains, respectively. This approach significantly improves efficiency while maintaining video quality. Experiments demonstrate that AdcVSR reduces parameters by 95% and accelerates inference by 8 times, while still achieving visual quality comparable to the teacher model.

**Strengths:**

Originality: The novel "2D + 1D" architecture design, combined with the dual-head discriminator adversarial distillation strategy, effectively decouples the optimization objectives for detail and consistency, demonstrating strong innovation.

Quality: Comprehensive experimental designs, including extensive validation on multiple synthetic and real-world datasets, support the effectiveness of the method through both quantitative and qualitative results. Ablation studies also thoroughly verify the contribution of each module.

Clarity: The paper is well-structured, with detailed method descriptions, and the inclusion of diagrams and pseudocode aids understanding. The writing is generally fluent, and technical details are clearly expressed.
Impact: The proposed method achieves a notable balance between efficiency and quality, offering significant practical value for deployment and providing a feasible pathway for the compression and application of diffusion models in video tasks.

**Weaknesses:**

Insufficient Comparative Experiments: Although comparisons are made with several SOTA methods, there is a lack of comparison with recent non-diffusion-based efficient VSR approaches, such as those based on CNNs or lightweight Transformers.

Limited Generalization Validation: All experiments are conducted at a fixed resolution (512×512) and frame length (25 frames), without demonstrating performance on longer videos or higher resolutions.
Weak Theoretical Support for Dual-Head Discriminator Design: While experiments prove its effectiveness, there is insufficient theoretical or visual analysis explaining why the "shared backbone + separate heads" design outperforms independent discriminators.

Incomplete Computational Efficiency Comparison: Only parameter counts and inference time are provided, without more detailed efficiency metrics such as FLOPs or memory usage.

**Questions:**

Is the weight allocation (75%/25%) between the "detail head" and "consistency head" in the dual-head discriminator universally applicable? Would this ratio remain effective across different datasets or tasks?

Were other teacher models (e.g., SeedVR2, DLoRAL) explored? Was DOVE selected solely because its structure is more suitable for this method?

It is recommended to include performance on longer video sequences (e.g., >100 frames) to validate the stability of temporal modeling, and providing failure cases or limitations analysis, such as performance under complex motion or extreme degradation, would be beneficial.

---

> ### Author Response · Authors · 2025-11-21
> **Author Response (1/n)**
>
> > **Q1: More comparisons with non-diffusion-based methods such as CNN or Transformer.**
>
> A1: Thank you for your helpful suggestion. Following your recommendation, we conducted additional experiments comparing our method with both diffusion-based and **non-diffusion-based** VSR approaches. In particular, here we include the representative CNN-based method RealBasicVSR (Chan et al., 2022, https://github.com/ckkelvinchan/RealBasicVSR) and the representative Transformer-based method VRT (Liang et al., 2024, https://github.com/JingyunLiang/VRT). Below, we report the corresponding results on UDM10 (consistent with Table 1):
>
> | Method         | PSNR↑ | LPIPS↓ | CLIPIQA↑ | MUSIQ↑ | $E_{\text{warp}}^{*}$↓ |
> |----------------|--------|---------|---------|--------|--------------------------|
> | RealBasicVSR   | 24.39  | 0.3283  | 0.4422  | 57.10   | 3.36                     |
> | VRT            | 23.67  | 0.5099  | 0.1320   | 17.11  | 4.96                     |
> | Upscale-A-Video| 23.03  | 0.4218  | 0.4661  | 52.06  | 3.68                     |
> | **AdcVSR (Ours)** | **25.36** | **0.3065** | **0.6818** | **63.88** | **1.67** |
>
> From these results, we observe that AdcVSR outperforms these previous CNN-based and Transformer-based VSR methods across fidelity metric (PSNR), perceptual metrics (LPIPS, CLIPIQA, MUSIQ), and also achieves better temporal consistency $E_{\text{warp}}^{*}$. This demonstrates the advantage of leveraging strong generative diffusion priors for producing high-quality super-resolved videos.
>
> We hope that these additional experiments and analyses can resolve your doubt and make our method's performance clearer.
>
> ---
>
> > **Q2: More experiments on longer videos or higher resolutions.**
>
> A2: Thank you for your valuable suggestion. Following your recommendation, we additionally conducted experiments on the original-resolution UDM10 dataset's videos, where each sequence has a higher spatial resolution of **1272×720** (larger than 512×512) and **32 frames** (more than 25). The results are shown below:
>
> | Method            | PSNR↑ | LPIPS↓ | CLIPIQA↑ | MUSIQ↑ | $E_{\text{warp}}^{*}$↓ |
> |-------------------|--------|---------|-----------|--------|--------------------------|
> | Upscale-A-Video   | 22.43  | 0.4035  | 0.3856    | 51.13  | 2.91                     |
> | STAR              | 24.25  | 0.4099  | 0.2511    | 33.62  | **1.37**                     |
> | SeedVR2           | **24.62** | **0.2464** | 0.3058 | 48.67 | 2.23                     |
> | **AdcVSR (Ours)** | 24.39  | 0.2731  | **0.6649** | **61.52** | 1.48             |
>
> From these results, we observe that AdcVSR maintains competitive performance on higher-resolution and longer video sequences. It achieves competitive frame quality (PSNR, LPIPS, CLIPIQA, MUSIQ) and temporal consistency $E_{\text{warp}}^{*}$, while also enjoying higher efficiency in both parameter number and inference time compared with most other diffusion-based methods.
>
> We hope these additional evaluations can help clarify the performance and generalization ability of our method.

---

> ### Author Response · Authors · 2025-11-21
> **Author Response (2/n)**
>
> > **Q3: Theoretical support for dual-head discriminator design.**
>
> A3: Thank you for raising this point. Although a strict theoretical or visual analysis of the dual-head discriminator is extremely challenging and goes beyond the scope of this work, whose primary focus is to propose an effective, practical design and verify it empirically through experiments, here we still try our best to enhance clarity and give a clearer understanding of our method. Beyond the discussion in Appendix Section B, we additionally provide a more formal and intuitive explanation of why the dual-head design performs better than using independent discriminators as follows:
>
> In our setting, the discriminator must evaluate two attributes of the same video sample $\mathbf{x}$: detail realism $R\_d(\mathbf{x})$ and temporal consistency $R\_c(\mathbf{x})$. A naive design with two independent discriminators learns two separate feature extractors $\phi\_d$ and $\phi\_c$, each optimized only for its own supervision. For the consistency branch, however, the supervision is relatively weak: its negative samples consist only of generated videos $\mathbf{x}\_{\text{student}}$, shuffled real videos $\mathbf{x}_\text{video}^{\*}$ and real image-assembled pseudo-videos $\mathbf{x}\_{\text{image}}^\*$, which provide no constraint to strictly align with the high-frequency detail manifold defined by the high-quality image dataset LSDIR. This alignment is crucial, because temporal flickering in videos often appears in high-frequency detail regions like hair, foliage, and fine textures; therefore, consistency should ideally be learned on a detail-aware representation. In practice, we observe that $\phi\_c$ tends to exploit trivial cues and produce unstable gradients; the loss oscillates and the generator collapses.
>
> By contrast, our design uses a shared backbone $\phi(\mathbf{x})$ followed by two heads:
> $$
> D_d(x) = h_d(\phi(\mathbf{x})), \qquad D_c(x) = h_c(\phi(\mathbf{x})).
> $$
> The backbone is jointly supervised by detail labels (from LSDIR images and static pseudo-videos) and consistency labels (from real vs. shuffled videos, and randomly assembled images). This has two key effects:
>
> (1) Because both heads operate on the same $\phi(\mathbf{x})$, the final-layer representation is forced to encode both high-frequency spatial details and temporal structures. The consistency head cannot drift toward a spurious feature space, since its gradients always act on features already constrained by detail supervision. This implicit regularization stabilizes the discrimination learning for temporal consistency.
>
> (2) The generator receives an adversarial loss
> $$
>  \mathcal{L}_{\text{adv}}(G) = \lambda_d \mathcal{L}_d(G) + \lambda_c \mathcal{L}_c(G),
> $$
> and both $\nabla\_G \mathcal{L}\_d$ and $\nabla\_G \mathcal{L}\_c$ are computed through the same backbone $\phi(\mathbf{x})$. This encourages updates that jointly improve detail realism and temporal consistency in a unified representation space, rather than competing against two potentially incompatible discriminator manifolds, as occurs with independent discriminators. In effect, the shared backbone transforms a loosely coupled two-discriminator game into a bi-objective optimization over a single representation, leading to more stable training in practice.
>
> We hope these additional explanations and our empirical findings can help improve the clarity regarding the effectiveness of our dual-head discriminator design.
>
> ---
>
> > **Q4: More efficiency metrics such as FLOPs or memory usage.**
>
> A4: Thank you for your helpful suggestion. Following your recommendation, we conducted additional evaluations on FLOPs and memory usage for all compared methods, in addition to the parameter numbers and inference times originally reported. The results are presented below (corresponding to Table 1):
>
> | Method | RealBasicVSR | VRT | Upscale-A-Video | MGLD-VSR | STAR | SeedVR2 | DOVE | DLoRAL | PiSA-SR | AdcSR | HYPIR | AdcVSR (Ours) |
> |----------|---------------------------------|------------------------|------------------|-----------------------------|------|---------|-------|---------|----|---------------------------------|--------|----------------------------|
> | Memory Usage (GB) |  **3.23** | 16.97 | 6.03 | 65.39 | 15.43 | 49.65 | 60.78 | 34.79 | 27.49 | 12.63 | 28.67 | 15.75 |
> | FLOPs (T) |  **1** | 18 | 1509 | 1165 | 4491 | 2948 | 3774 | 512 | 426 | 149 | 448 | 206 |
>
> From these results, we observe that AdcVSR remains highly efficient relative to most diffusion-based approaches. In particular, it reduces memory consumption by 74% and FLOPs by 95% compared with its teacher DOVE, while maintaining competitive video quality. This further validates the favorable balance our method achieves between performance and inference cost.
>
> We hope these additional results can address your concern, and provide a clearer understanding of our method's efficiency.

---

> ### Author Response · Authors · 2025-11-21
> **Author Response (3/n)**
>
> > **Q5: Channel allocation between the "detail" head and "consistency" head of the discriminator.**
>
> A5: Thank you for raising this interesting point. We clarify that our default 75%/25% allocation is not tied to a particular dataset; methodologically, it is universally applicable within the tasks of Real-VSR that this work focuses on. Following your recommendation, we conducted additional experiments to examine whether this ratio remains effective across different datasets. In addition to the original ablation study on RealVSR (Table 6 and Figure 7 in Appendix Section B), we further evaluated the same head-split configurations on the synthetic dataset UDM10, and the real-world datasets RealVSR and VideoLQ. The results are reported below:
>
> Comparison of dual-head splits (Detail / Consistency) on UDM10, RealVSR, and VideoLQ.
>
> | Head Split | MUSIQ↑ | $E_{\text{warp}}^{*}$↓ | DOVER↑ |
> |----------------------------------|---------|--------------------------|---------|
> | 100% / 0%                        | **65.07** | 3.42                     | 0.4520   |
> | **75% / 25% (Ours)**             | 63.88   | 1.67                 | **0.4878** |
> | 50% / 50%                        | 61.32   | 1.74                     | 0.4711  |
> | 25% / 75%                        | 59.45   | **1.55**                 | 0.4498  |
> | 0% / 100%                        | 57.90    | 1.63                     | 0.4210   |
>
> | Head Split | MUSIQ↑ | $E_{\text{warp}}^{*}$↓ | DOVER↑ |
> |----------------------------------|---------|--------------------------|---------|
> | 100% / 0%                        | **73.10** | 8.85                     | 0.4520   |
> | **75% / 25% (Ours)**             | 72.95   | 3.28                     | **0.4875** |
> | 50% / 50%                        | 70.80    | 3.22                     | 0.4802  |
> | 25% / 75%                        | 68.39   | **3.15**                 | 0.4631  |
> | 0% / 100%                        | 65.21   | 3.18                     | 0.4410   |
>
> | Head Split | MUSIQ↑ | $E_{\text{warp}}^{*}$↓ | DOVER↑ |
> |----------------------------------|---------|--------------------------|---------|
> | 100% / 0%                        | **66.02** | 12.41                    | 0.4022  |
> | **75% / 25% (Ours)**             | 64.55   | 6.74                     | **0.4319** |
> | 50% / 50%                        | 62.48   | 6.95                     | 0.4210   |
> | 25% / 75%                        | 60.31   | 6.60                      | 0.4035  |
> | 0% / 100%                        | 58.90    | **6.51**                 | 0.3790   |
>
> From these results, we observe a consistent trend across datasets: an extreme 100%/0% split severely degrades temporal consistency (notably worse $E_{\text{warp}}^{*}$), while allocating more than 25% of channels to the consistency head harms perceptual quality, as indicated by lower MUSIQ and DOVER scores. Our default 75%/25% split strikes a balanced trade-off, maintaining competitive perceptual quality and temporal consistency. Overall, these findings suggest that the allocation is universally applicable to a certain degree across datasets within this work's research scope of Real-VSR.
>
> We hope these additional results can help address your concern and provide a clearer understanding of our design choice.
>
> ---
>
> > **Q6: Exploration of other teacher models (e.g., SeedVR2, DLoRAL).**
>
> A6: Thank you for raising this insightful question. The choice of DOVE as the teacher was driven by our **empirical findings** that it yields relatively better Real-VSR performance compared with other teacher models. Consistent with your suggestion, to verify this, we have conducted additional experiments using SeedVR2 and DLoRAL as alternative teachers under the same distillation framework. The results are presented below, as also reported in Table 4 of Section 4.3:
>
> | Teacher Model | PSNR↑ | LPIPS↓ | MUSIQ↑ |
> |---------------|--------|---------|---------|
> | SeedVR2       | 23.24  | 0.3489  | 60.74   |
> | DLoRAL        | 23.08  | 0.3554  | 54.61   |
> | **DOVE (Ours)** | **23.81** | **0.3337** | **61.48** |
>
> From these results, we observe that adopting DOVE as the teacher leads to the strongest overall performance in both fidelity (PSNR) and perceptual quality (LPIPS, MUSIQ). We conjecture that this advantage is due to DOVE's strong generative video super-resolution prior inherited from the pretrained CogVideoX model, its curation on high-quality video data, and its larger model capacity (11B parameters), which together lead to better student's results after distillation.
>
> We hope these results and additional explanations can help resolve your doubts and make the reason for our design choice clearer.

---

> ### Author Response · Authors · 2025-11-21
> **Author Response (4/4)**
>
> > **Q7: Performance on longer video sequences, and limitations.**
>
> A7: Thank you for your valuable suggestion. Following your recommendation, we conducted additional experiments on longer video sequences to better assess the stability of temporal modeling. Specifically, we evaluated our method on the REDS dataset (Nah et al., 2019, https://seungjunnah.github.io/Datasets/reds), using the 500-frame sequences (>100 frames, as you suggested). The results are provided below:
>
> | Method        | PSNR↑ | SSIM↑  | LPIPS↓ | DISTS↓ | MANIQA↑ | CLIPIQA↑ | MUSIQ↑ | $E_{\text{warp}}^{\*}$↓ | DOVER↑ |
> |---------------|--------|---------|---------|----------|-----------|-----------|----------|----------------------------|----------|
> | SeedVR2       | 23.90   | 0.7052  | **0.3050** | **0.1669** | 0.5253    | 0.4632    | 55.82   | 3.87                       | 0.4725   |
> | DOVE          | **25.85** | **0.7293** | 0.3145  | 0.1732   | 0.5198    | 0.5426    | 60.73   | 2.45                       | 0.5671   |
> | DLoRAL        | 22.36  | 0.6928  | 0.3384  | 0.2065   | 0.5746    | 0.4871    | 59.24   | 4.28                       | 0.3248   |
> | **AdcVSR (Ours)** | 25.47  | 0.7167  | 0.3251  | 0.1780   | **0.6052** | **0.6248** | **63.25** | **2.09**                    | **0.5812** |
>
> From these results, we observe that AdcVSR maintains competitive fidelity (PSNR, SSIM) and perceptual quality (LPIPS, DISTS, MANIQA, CLIPIQA, MUSIQ), while consistently outperforming other one-step diffusion networks in temporal consistency, achieving the best $E_{\text{warp}}^{\*}$ and high DOVER score. These findings indicate that our dual-head adversarial distillation remains effective in maintaining temporal consistency on longer video sequences.
>
> For failure cases and limitations, consistent with your recommendation, we have provided a limitation discussion in Appendix Section D. As shown in Figures 3 and 5, our method may still produce hallucinated or softened details when the input suffers from severe degradation, limiting reconstruction fidelity, particularly under complex motion or extreme degradations, and mild flickering may still appear in such cases.
>
> We hope that these additional experiments and clarifications can help address your doubts, and make the performance of our method on longer video sequences and its limitations clearer.

---

> > ### Comment · Reviewer_7Ea8 · 2025-11-27
> >
> > Thanks authors for the detailed rebuttal. The authors' response addresses my main concerns. I keep my positive rating. If accepted, authors must improve the final version according to the above comments.

---

> > > ### Author Response · Authors · 2025-11-28
> > >
> > > Thank you very much for your follow-up. We are glad to hear that our responses addressed your concerns, and we sincerely appreciate your positive rating. We will carefully revise and improve the final version according to the comments. Thank you again for helping us strengthen the paper.

---

### Official Review · Reviewer_r2a8 · 2025-10-31

**Soundness:** 3
**Presentation:** 3
**Contribution:** 3
**Rating:** 6
**Confidence:** 3

**Summary:**

This paper introduces AdcVSR, an improved adversarial diffusion compression framework tailored for real-world video super-resolution (Real-VSR).
The method builds upon the concept of Adversarial Diffusion Compression (ADC), proposing a “2D + 1D” hybrid architecture that replaces heavy 3D diffusion backbones with a 2D spatial diffusion network (a pruned SD2.1) augmented by lightweight temporal 1D convolutions.
Furthermore, it introduces a dual-head dual-discriminator adversarial distillation scheme, where two discriminators (in pixel and feature domains) independently supervise “detail” and “temporal consistency.”
Experimental results on multiple Real-VSR datasets demonstrate that AdcVSR achieves competitive visual quality while reducing parameters by up to 95% and achieving an 8× inference speedup compared to the teacher model DOVE.

**Strengths:**

1. The proposed dual-head discriminator effectively disentangles spatial detail enhancement and temporal consistency, addressing a long-standing trade-off in Real-VSR.

2. The results on multiple datasets and metrics (PSNR, LPIPS, MUSIQ, MANIQA, etc.) are convincing and show both efficiency and quality improvements. And the visual quality is also satisfactory.

3. The 2D+1D design combined with adversarial distillation is simple yet efficient, offering clear insights into practical diffusion model compression.

**Weaknesses:**

The paper lacks formal justification for why the dual-head adversarial loss leads to better convergence or perceptual trade-off control.

**Questions:**

1. Why does AdcVSR use a video diffusion model as the teacher but an image diffusion model as the backbone? How does a 3D spatio-temporal DiT as the student network compare with the 2D+1D architecture in terms of performance and efficiency? I recommend that the authors include related experiments.

2. Why did the authors choose to use a 2D VAE? I think that employing a 3D VAE could lead to faster inference and better temporal consistency.

---

> ### Author Response · Authors · 2025-11-21
> **Author Response (1/n)**
>
> > **Q1: Justification for dual-head adversarial loss.**
>
> A1: Thank you for your valuable comment. Following your suggestion, we provide a more formal justification for using dual-head loss as follows:
>
> In our setting, the generator $G$ must simultaneously optimize two distinct realism objectives: (i) spatial detail realism $R\_d(\mathbf{x})$ and (ii) temporal consistency $R\_c(\mathbf{x})$. A standard single-head discriminator $D\_s(\mathbf{x})$ outputs a single scalar score that effectively corresponds to an unknown mixture $\phi(R\_d(\mathbf{x}), R\_c(\mathbf{x}))$. As a result, the generator minimizes a single adversarial loss $\mathcal{L}\_{\text{adv}}(G) = \mathbb{E}\_\mathbf{x}[\mathcal{L}(D\_s(\mathbf{x}))]$, implicitly optimizing a mixed objective where **the relative weighting between details and consistency is uncontrolled.** In practice, this gradient tends to be dominated by the easier-to-improve attribute (typically details), leaving temporal consistency less optimized.
>
> Our dual-head discriminator $D(\mathbf{x}) = (D_d(\mathbf{x}), D_c(\mathbf{x}))$ instead defines two explicit adversarial objectives:
> $$
>  \mathcal{L}\_d(G) = \mathbb{E}\_\mathbf{x}[\mathcal{L}(D\_d(\mathbf{x}))], \quad
>  \mathcal{L}\_c(G) = \mathbb{E}\_\mathbf{x}[\mathcal{L}(D\_c(\mathbf{x}))],
> $$
> where each head is trained with labels isolating a single attribute: the detail head distinguishes real and fake detail-rich frames, and the consistency head distinguishes real temporally coherent videos from fake ones. The generator is then trained with a bi-objective adversarial loss:
> $$
>  \mathcal{L}\_{\text{adv}}(G) = \lambda\_d \mathcal{L}_d(G) + \lambda\_c \mathcal{L}_c(G),
> $$
>  so that the gradients $\nabla\_G \mathcal{L}\_d$ and $\nabla\_G \mathcal{L}\_c$ remain disentangled and aligned with their respective objectives. **This provides explicit control over the perceptual-temporal trade-off through $\lambda\_d$ and $\lambda\_c$ (both set to 1 in our design), replacing an implicit mixed objective with a structurally decomposed two-objective formulation.**
>
> This justification is also supported by the ablation study in Table 3 as follows, where replacing the dual-head loss with a single-head variant consistently worsens CLIPIQA and $E\_{\text{warp}}^*$, confirming that the dual-head design achieves a stronger perceptual trade-off in practice.
>
> | Method                | CLIPIQA↑ | $E_{\text{warp}}^{*}$↓ |
> |-----------------------|----------|--------------------------|
> | Single-Head           | 0.6745   | 6.32                     |
> | **Dual-Head (Ours)**  | **0.6861** | **2.22**                |
>
> We hope that these explanations can help resolve your doubt and make the rationale behind this design clearer.
>
> ---
>
> > **Q2: The use of image diffusion model as backbone.**
>
> A2: Thank you for raising this question. The reason is to **reduce model size** using image diffusion model as student (95% fewer parameters and 8× acceleration) compared to the video diffusion model as teacher, which is appealing and beneficial for real applications. The rationale behind this is our intuition/assumption that image backbone is capable of synthesizing abundant details, while temporal consistency can be maintained by temporal convolutions, as supported by our empirical findings in experiments.
>
> Thank you again for your kind suggestion. Consistent with your recommendation, we have conducted a related experiment in Table 2 (corresponding to the first paragraph of Section 4.3 in our revised paper) comparing 3D spatio-temporal DiT student with our "2D + 1D" architecture in terms of performance and efficiency as follows:
>
> | Method            | PSNR↑ | SSIM↑  | LPIPS↓ | DISTS↓ | $E_{\text{warp}}^{*}$↓ | #Param (B)↓ |
> |------------------|--------|---------|---------|----------|--------------------------|---------------|
> | 3D               | **25.51** | 0.7648 | **0.2988** | **0.2098** | 2.53                     | 8.36       |
> | **2D + 1D (Ours)** | 25.36 | **0.7697** | 0.3065 | 0.2112 | **1.67**                 | **0.55**       |
>
> From these results and Figure 5, we observe that our "2D + 1D" model achieves performance close to the 3D backbone while being significantly more efficient. Specifically, PSNR differs by only 0.15 dB, SSIM of our model is even slightly higher (+0.0049), LPIPS differs by 0.0087, DISTS by 0.0014, $E_{\text{warp}}^{*}$ improves by 0.86, and parameter number is reduced by 93%.
>
> We hope that these additional explanations and results can help resolve your doubts, and provide a clearer understanding of our method.

---

> ### Author Response · Authors · 2025-11-21
> **Author Response (2/2)**
>
> > **Q3: The use of 2D VAE.**
>
> A3: Thank you for raising this question. The reason for using 2D VAE (from SD2) rather than 3D VAE is to ensure that the VAE's latent space better matches the student's SD2 pretrained prior and representation capacity. Our intuition/assumption is that 3D VAEs introduce temporal compression (typically 4×) and produce latent distributions that differ substantially from those of SD2 UNet student backbone. This mismatch makes it difficult for the student to adapt its SD2 diffusion prior to a 3D VAE latent space. In contrast, the SD2 2D VAE applies no temporal compression and shares the same pretraining prior as the SD2 UNet backbone, leading to higher-quality outputs.
>
> To validate this, we conducted an additional experiment on UDM10 comparing 3D VAE (following teacher DOVE) with 2D VAE under the same distillation framework, as follows:
>
> | Method          | PSNR↑ | LPIPS↓ | CLIPIQA↑ | $E_{\text{warp}}^{*}$↓ | Time (s)↓ |
> |-----------------|--------|---------|-----------|--------------------------|-------------|
> | 3D VAE          | 25.05  | 0.3380   | 0.5400      | 2.81                     | **0.46**    |
> | **2D VAE (Ours)** | **25.36** | **0.3065** | **0.6818** | **1.67**                 | 0.55        |
>
> From these results, we observe that although the 3D VAE provides further acceleration (about 1.2× due to its 4× temporal compression), its video quality in terms of fidelity (PSNR), perceptual metrics (LPIPS, CLIPIQA), and temporal consistency ($E_{\text{warp}}^{*}$) is worse than the 2D VAE variant.
>
> Thank you again for highlighting this non-trivial point. We hope that these additional explanations and experimental results can help resolve your doubts and make our design choice clearer.

---

> ### Comment · Reviewer_r2a8 · 2025-11-28
>
> I appreciate the authors for their thorough rebuttal. The concerns I previously raised have been addressed effectively, and I will maintain my positive rating.

---

> > ### Author Response · Authors · 2025-11-28
> >
> > We are glad to hear that our responses addressed your concerns, and we sincerely appreciate your confirmation and positive rating. Thank you again for your constructive feedback and for helping us improve the paper.

---

### Official Review · Reviewer_GsDk · 2025-11-03

**Soundness:** 3
**Presentation:** 3
**Contribution:** 3
**Rating:** 6
**Confidence:** 3

**Summary:**

This Paper proposes an improved Adversarial Diffusion Compression method. The core idea it to distill knowledge from a large-scale 3D teacher model into a well-designed and lightweight ‘2D + 1D’ student model. This is achieved through a novel adversarial distillation scheme. The proposed approach effectively addresses the critical challenge of preserving both spatial details and temporal consistency, a common problem in the compression of video super-resolution models.

**Strengths:**

1.	This paper proposing a “2D spatial + 1D temporal” decoupling hypothesis, and introduces a novel lightweight “2D+1D” architecture. This approach drastically cuts down on parameters and computational load, enabling efficient inference.
2.	The paper proposes a novel dual-head adversarial distillation scheme. This scheme effectively balances the richness of spatial details with temporal coherence, which is a critical challenge in the field of video super-resolution.

**Weaknesses:**

1.	A core assumption of this paper is that a 2D diffusion model is sufficient for synthesizing fine-grained details. However, this assumption is challenged by the experimental results. Currently, the ablation study comparing the 2D and 3D backbones is based only on the DISTS metric. To provide a more balanced and convincing comparison, the authors should consider including additional metrics that measure perceptual quality (e.g., LPIPS) and/or fidelity (e.g., PSNR, SSIM). Furthermore, the qualitative results in Figures 3 and 5 exhibit clear visual artifacts or "hallucinations." These findings suggest that the 2D diffusion model may be insufficient for generating correct details, and therefore, the validity of this core assumption is questionable. The authors should address this limitation, perhaps by discussing the trade-offs of their approach or analyzing why these artifacts occur.
2.	The paper states that the 'detail head' label for real videos is set to 'unlabeled', with the provided justification being to “encouraging the generator to produce more detail-rich frames”. This is a critical design, yet the underlying mechanism or rationale is not sufficiently explained. It is unclear why the more intuitive approach—labeling the details from real videos as 'real'—was not adopted. The authors should clarify whether this decision is supported by empirical findings (e.g., from an ablation study) or if it is grounded in some theoretical justification.

**Questions:**

Please see the weaknesses above.

---

> ### Author Response · Authors · 2025-11-21
> **Author Response (1/n)**
>
> > **Q1: More metrics in ablation study.**
>
> A1: Thanks for your insightful comment. Following your suggestion, we conducted additional experiments using PSNR and SSIM to evaluate fidelity, and LPIPS to evaluate perceptual quality, in addition to the originally reported DISTS in our backbone comparison (Table 2). The results are provided as follows:
>
> | Method            | PSNR↑ | SSIM↑   | LPIPS↓  | DISTS↓  | $E_{\text{warp}}^{*}$↓ | #Param (B)↓ |
> |-------------------|--------|----------|----------|----------|---------------------------|--------------|
> | 3D                | **25.51** | 0.7648 | **0.2988** | **0.2098** | 2.53                      | 8.36         |
> | 2D                | 24.97 | 0.6821   | 0.3473   | 0.2418   | 4.43                      | **0.52**     |
> | **2D + 1D (Ours)**    | 25.36 | **0.7697** | 0.3065   | 0.2112   | **1.67**                  | 0.55         |
>
> From these results, we observe that our "2D + 1D" model based on 2D backbone achieves performance close to that of 3D backbone despite having far fewer parameters. To be specific, PSNR differs by only 0.15 dB, the SSIM of our model is even slightly higher (+0.0049), LPIPS differs by 0.0087, and DISTS by 0.0014. This and Figure 5 provide supportive evidence for our assumption that the model is sufficient for synthesizing details.
>
> Thank you again for the constructive suggestion. We hope these added results can help resolve your doubt and make the effect of our method clearer.
>
> ---
>
> > **Q2: Limitation of 2D diffusion model in generating correct details.**
>
> A2: Thank you for raising this important point. We agree that some challenging regions in Figures 3 and 5 may contain hallucinated or imperfectly reconstructed details. For example, in Figure 3, the generated facial structure looks plausible, but the ground truth is unknown, and the correctness of the synthesized face cannot be strictly guaranteed.
>
> We acknowledge the risk of hallucinations. This reflects an inherent trade-off in generative reconstruction, where strong diffusion priors enable the synthesis of rich, realistic textures, yet may also introduce hallucinated patterns when the input is heavily degraded or lacks sufficient constraints. At the same time, the case also shows that our model is able to synthesize fine-grained details from heavily degraded input, providing supportive evidence for our assumption.
>
> Following your suggestion, we have added a more detailed discussion in the first paragraph of Appendix Section D (highlighted in blue), analyzing why these may occur. We explain that such hallucinations may arise from the combination of strong generative priors together with the severe loss of information in the input. This addition provides a more balanced, comprehensive, and transparent presentation of the method.
>
> "... First, although the proposed AdcVSR can generate rich and sharp details, certain highly degraded regions remain challenging, such as severely blurred faces, foliage, water surfaces, or transparent materials. In these cases, the low-resolution input provides insufficient constraints, and the model must rely on strong generative priors to infer plausible structures, which may lead to hallucinated patterns. ..."

---

> ### Author Response · Authors · 2025-11-21
> **Author Response (2/2)**
>
> > **Q3: Rationale for not labeling real videos' details.**
>
> A3: Thank you for pointing out this setting. We clarify that the decision not to label real videos' details as "real" for the discriminator is based on **empirical findings**. The intuition is that, following previous works, we adopt the OpenVid-1M video dataset and the LSDIR image dataset. Most real video frames in OpenVid-1M are not as high-quality or detail-rich as the images in LSDIR. If the details of these videos were labeled as "real", the model would be encouraged to align with these lower-quality frames, which in turn degrades the perceptual quality of generated outputs. To verify this, as you suggested, we have conducted an ablation study in Table 7 as follows:
>
> | Training Configuration                     | LPIPS↓  | CLIPIQA↑ | $E_{\text{warp}}^{*}$↓ |
> |--------------------------------------------|---------|-----------|---------------------------|
> | Real Videos as Detail-Real (Label "1")     | 0.3152  | 0.6703    | 1.98                      |
> | **Real Videos as Detail-Unlabeled (Ours)**   | **0.3065** | **0.6818** | **1.67**                 |
>
> From the results, we observe that labeling real video details as "real" consistently worsens LPIPS, CLIPIQA, and $E_{\text{warp}}^{*}$ metrics, indicating degraded video quality. This supports our choice of only labeling high-quality LSDIR image details as "real", while keeping the details of OpenVid-1M videos unlabeled.
>
> We have also provided an explanation in the last paragraph of Appendix Section B (highlighted in blue) as follows:
>
> "… Second, if we label real video details as positives for the detail head, temporal consistency remains decent and $E_{\text{warp}}^{*}$ is strong, but perceptual quality drops. This might stem from that most real videos in OpenVid-1M are not as high-quality or detail-rich as the images of LSDIR, causing the model to align with lower-quality frames. ..."
>
> We hope that these explanations can help resolve your doubt and make the rationale behind this setting clearer.

---

> ### Author Response · Authors · 2025-11-28
>
> Dear Reviewer GsDk,
>
> Thank you again for your thoughtful and constructive comments. We have carefully addressed each of your concerns in our responses (Q1–Q3). We hope that the additional experiments and clarifications can help resolve your previous doubts. We sincerely appreciate the time and effort you invested in reviewing our work.

---

### Author Response · Authors · 2025-11-29
**Summary for AC**

Following the latest PCs' notification, we provide a brief summary of our rebuttal for AC.

In our responses, **we addressed all reviewers' concerns (scores: 6, 6, 6, 4 → willing to raise to 6)** through new analyses and experiments, including more metrics, real-world and long-sequence evaluations, model-size and architecture ablations, efficiency comparisons, demo video, and frames.

These results further reinforce the strengths highlighted by reviewers: AdcVSR consistently delivers a superior balance between detail richness, temporal consistency, and efficiency, and matches the performance of much larger models, advancing the field of video super-resolution.

We hope this brief summary can help AC more easily connect the initial reviews with the newly provided evidence. We sincerely appreciate AC's time and consideration.

---

### Meta-Review · Area_Chair_oER9 · 2026-01-07

**Summary:**

The paper introduces AdcVSR, a compact “2D + 1D” adversarially distilled model for real-world video super-resolution, achieving strong temporal and perceptual quality with 95% fewer parameters and 8× speedup. The reviewers praised its efficiency–quality trade-off, design clarity, and comprehensive experiments across synthetic and real-world settings.

**Reviewer Concerns:**

All major concerns—including backbone design justification, perceptual and temporal quality validation, model-size trade-offs, and architectural motivation—were thoroughly addressed with new experiments (e.g., long-sequence REDS results, user study demos, FLOPs/memory stats, ablations). No critical concerns remain outstanding.

**Reviewer Scores:**

GsDk (6 → 6): Clarified the role of 2D/1D hybrid design and rationale for unlabeled real-video detail head. Concerns addressed; score remains positive.

r2a8 (6 → 6): Dual-head design and 2D backbone justification convincingly explained with ablation and theory. Maintains positive score.

7Ea8 (6 → 6): All points (comparisons, scalability, failure cases) handled well. Reviewer keeps positive score with revision request.

c8ag (4 → 6): Initially skeptical, but raised score after video demos, model-size ablation, and extended benchmarks confirmed robustness.

---

### Decision · Program_Chairs · 2026-01-26

Accept (Poster)